# FEDERATED GENERATIVE LEARNING WITH FOUNDATION MODELS

## ABSTRACT

Existing federated learning solutions focus on transmitting features, parameters, or gradients between clients and server, which suffer from serious low-efficiency and privacy-leakage problems. Thanks to the emerging foundation generative models, we propose a novel federated learning framework, namely *Federated Generative Learning*. In this framework, each client can create text prompts that are tailored to their local data, based on its features, and then send them to the server. Given the received prompts, the informative training data can be remotely synthesized on the server using foundation generative models. This new framework offers several advantages, including enhanced communication efficiency, improved resilience to distribution shift, significant performance gains, and enhanced privacy protection. We validate these benefits through extensive experiments conducted on ImageNet and DomainNet datasets, *e.g.*, on ImageNet100 dataset, with a highly skewed data distribution, our method outperforms FedAvg by **12%** in a single communication round. Moreover, our approach only requires **229 Bytes** prompts for communication, while FedAvg necessitates the transmission of **42.7 MB** parameters.

## 1 INTRODUCTION

Recently, significant progress has been achieved in many learning fields by scaling up to large models, *i.e.*, BERT Devlin et al. (2018), GPT3 Brown et al. (2020), ViT Dosovitskiy et al. (2020), CLIP Radford et al. (2021), Stable Diffusion Rombach et al. (2022a), and web-scale datasets *i.e.*, YFCC100M Thomee et al. (2016), CC-12M Changpinyo et al. (2021), LAION-5B Schuhmann et al. (2022a). Typically, large models are first pre-trained with massive low-quality web data for basic capability, then finetuned with a small number of high-quality data, especially manually labeled data, for evoking the desired capability. Although web data are easily accessible, high-quality training data remains scarce due to the fact that high-quality datasets are typically private or unsuitable for public release. For example, the process of labeling medical data is often costly, and the release of such data is sensitive due to safety and privacy concerns. Furthermore, raw data itself are often considered a valuable asset for numerous companies, rendering its acquisition impractical. Consequently, there is a pressing need for collaborative machine learning Gong et al. (2022); Nguyen & Thai (2022); Mothukuri et al. (2021) that is both efficient and privacy-preserving.

Recently, Federated Learning (FL) McMahan et al. (2017) has been gaining a lot of attention as a potential way to protect user privacy in distributed machine learning. When it comes to practical applications, FL systems face a few challenges that impede their real-world implementation:

1. ***High communication cost***. Current FL solutions require the transmission of model parameters or gradients between clients and the server McMahan et al. (2017); Zhao et al. (2018). However, in the era of large models, these parameters are often in the billions or trillions, making their communication costly and even prohibitively expensive.

2. ***Data heterogeneity***. In FL, a fundamental challenge arises from the presence of statistical heterogeneity among local data distributions across distinct clients. This leads to the problem of objective inconsistency for FL methods based on model average, as the global model converges to a stationary point of a mismatched objective Wang et al. (2020); Zhang et al. (2022b), causing remarkable performance degradation.

3. ***Privacy and security risks***. The main focus of FL systems is to guarantee data privacy and security. Existing techniques usually involve the transmission of model parameters or gradients McMahan et al. (2017) or artificial data Zhou et al. (2020). Nevertheless, these approaches are vulnerable to considerable privacy leakage Zhao et al. (2020).

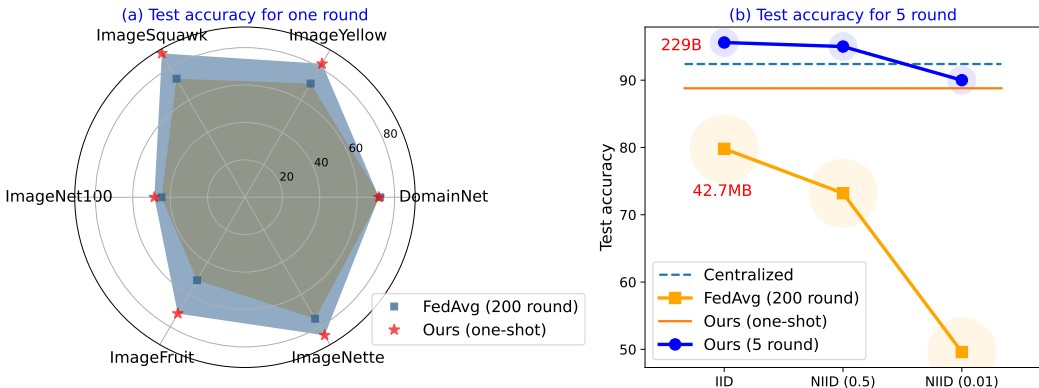

Figure 1: **Left:** We compared the performance of FedAvg and our method in a single round on five ImageNet subsets and a DomainNet subset. **Right:** Our results surpass FedAvg in various data distributions on the ImageSquawk dataset for 5 rounds. The results are even superior to centralized training. The circles indicate the transmission cost in each communication round, measured in bytes.

Recent advances in foundation generative models, *i.e.*, Stable diffusion Rombach et al. (2022b), DALL-E2 Ramesh et al. (2022), Imagen Saharia et al. (2022), and GLIDE Nichol et al. (2021), have provided a high-quality text conditional image synthesis which can be used to train models. These foundation generative models have been applied to various fields, including Medical Imaging Zhu et al. (2023); Rahman et al. (2023), Computer Vision Blattmann et al. (2023); Ruiz et al. (2023), Speech Liu et al. (2023); Levkovitch et al. (2022), and have achieved remarkable outcomes.

In this paper, we propose a novel framework called *Federated Generative Learning* (**FGL**), which leverages powerful foundation generative models, *e.g.*, Stable Diffusion Rombach et al. (2022a), to synthesize high-quality training data on the server based on the prompts collected from clients. We propose two customized prompt generation methods based on the characteristics of the client's data: class-level prompt and instance-level prompt. These methods enable clients to provide informative prompts that capture the unique features of their data. Once all the prompts are collected from the clients, the server performs prompt aggregation and then synthesizes a high-quality substitute training dataset. This dataset serves as a proxy for the clients' private data and can be used to train a global model in a centralized manner. As demonstrated in Figure 1(a), in a single round, our trained model outperforms traditional FL methods that require hundreds of communication rounds, both on five ImageNet subsets and one subset of DomainNet. In summary, there are multiple benefits of FGL:

1. *Low Communication Cost*. Compared to previous methods that rely on multi-round communication of weights or gradients, our method requires only one or a few communication rounds (*e.g.*, 5 rounds) between clients and the server. As shown in Figure 1(b), during the initial communication between any client and the server, the parameter transmission in FedAvg requires 42.7MB for the small network ResNet18, while our method only sends 229 Bytes prompts. This advantage will be enlarged when training large models with billions of parameters and implementing federated continual learning in which requires more communication rounds for learning from new data.

2. *Robust to Data Heterogeneity*. Since our method only requires clients to upload prompts corresponding to their local training data, it allows the server to collect prompts from all clients and synthesize all training data in one communication round. As a result, our method exhibits insensitivity to data distribution. Figure 1(b) compares our method to FedAvg under three different data distributions. Ours (one-shot), consistently achieves good performance across all distributions. Furthermore, our method can significantly improve the performance and even outperform the centralized training by implementing five-round communication.

3. *Better privacy-preserving*: Traditional FL involves the transmission of model parameters or gradients between clients and server. However, once these model parameters are leaked, the attackers can carry out model extraction attacks Li et al. (2023b), member inference attacks Li et al. (2023a), and other malicious activities Zhang et al. (2023b); Zhu et al. (2019), posing significant security threats to the FL system Nguyen & Thai (2022); Liu et al. (2022). In contrast, foundation generative model based FL can better protect data privacy by transmitting prompts. Our method can implement FL in few round communication which significantly reduces the privacy leakage risk. In Section 5.4, we conduct a thorough privacy analysis on two aspects: 1) whether the synthetic data incorporates private data information and 2) whether the model trained on the synthetic data is resilient against privacy attacks.

## 2 RELATED WORK

**Foundation Generative Models.** Large generative models, such as Stable Diffusion Rombach et al. (2022b), DALL-E2 Ramesh et al. (2022), Imagen Saharia et al. (2022), and GLIDE Nichol et al. (2021), have recently emerged as an off-the-shelf tool for high-quality and real-looking image generation conditioned on text prompts. A few works have explored the usage of synthetic images as training data. For example, He et al. He et al. (2022) show that synthetic data generated by diffusion models can improve pretraining, zero-shot, and few-shot image classification performance. Li et al. Li et al. (2023d) demonstrate that synthetic data generated by conditional diffusion models can be used for knowledge distillation without access to the original data. Zhou Zhou et al. (2023) synthesize better images for model training with stable diffusion by implementing diffusion inversion. Although good performance has been achieved, inverting every image is expensive and will leak privacy.

**Foundation Models in FL.** The foundation generative models are still under-explored in federated learning, though there exist a few related works that study foundation models in federated learning. The most similar one is Yang et al. (2023), which takes advantage of the diffusion model in the server to synthesize training samples that complied to the distributions of domain-specific features from clients. Yu et al. Yu et al. (2023) introduce federated learning into foundation model training for training foundation models collaboratively with private data. Like traditional FL methods, they also transmit model parameters between servers and clients. Based on the shared CLIP model, Guo et al. Guo et al. (2022) transmit the small number of updated parameters of the prompt learner from clients to the server to reduce the communication cost. Our study is different from them in multiple aspects: 1) Their methods transmit features or parameters, while our federated generative learning framework transmits prompts which require less communication while are better at privacy-preserving. 2) Our framework is based on shared foundation generative models, while theirs are based on the shared foundation classification models.

## 3 PRELIMINARIES

### 3.1 TEXT-TO-IMAGE GENERATIVE MODELS

Text-to-image generative models, such as Stable Diffusion Rombach et al. (2022b), DALL-E2 Ramesh et al. (2022), Imagen Saharia et al. (2022), have demonstrated the remarkable ability to generate stunning images from natural language descriptions. Given a pre-trained generative model $G$, such as the stable diffusion model, users can simply provide a text prompt, which enables the synthesis of high-quality images. Specifically, during the inference phase, a new image latent $z_0$ is generated by iteratively denoising a random noise vector using a conditioning vector, *e.g.*, text prompt embedding $p$. The latent code is subsequently transformed into an image by employing the pre-trained decoder, resulting in $x' = G(z_0, p)$.

In this paper, we only use off-the-shelf pre-trained diffusion model for inference process, *i.e.*, image generation, without conducting any training.

### 3.2 FEDERATED LEARNING

In the setting, we have $K$ clients with their private datasets $\mathcal{D}_k = \{(x_i, y_i)\}_{i=1}^{N_k}$, where $x_i$ is the training image, $y_i$ is its label, $\mathcal{Y}_k$ is the label set, and $N_k$ is the number of training samples in $k$-th client. Note that the label sets of different clients may be different. The objective of the federated learning framework is to learn a model parameterized with $\theta$ in the server that minimizes the loss on training data of all clients without access to original data:

$$\min_{\theta} \frac{1}{K} \sum_{k=1}^{K} \mathbb{E}_{x \sim \mathcal{D}_k}[l_k(\theta; x)], \tag{1}$$

where, $l_k$ is the loss function, *i.e.*, cross-entropy loss, for the $k$-th client.

## 4 FEDERATED GENERATIVE LEARNING

The overall framework of the proposed *Federated Generative Learning* framework is illustrated in Figure 2. Unlike traditional federated learning methods that transmit feature, parameters or gradients,

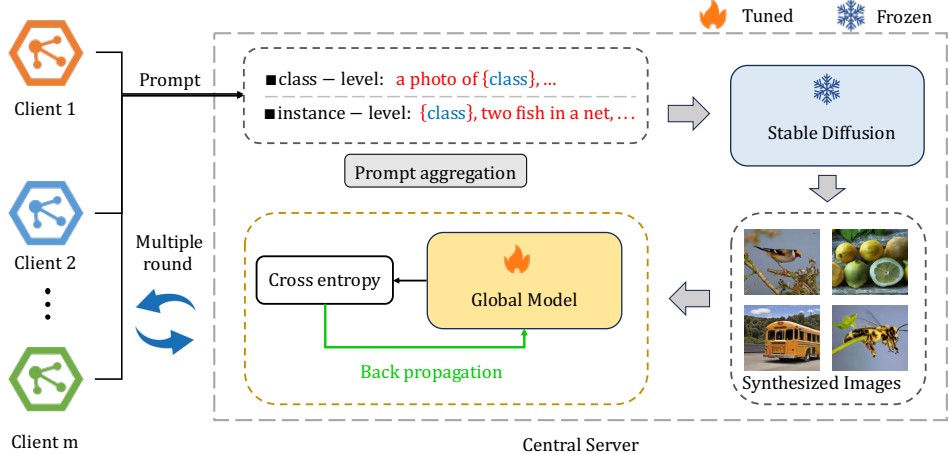

Figure 2: Training pipeline of Federated Generative Learning. Firstly, the prompts from clients are uploaded and then aggregated on the server. Then, stable diffusion is used to generate synthetic data to train the global model. Then, the updated model weights are distributed to the clients.

our approach transmits prompts corresponding to the private data in clients to the server, thus being better privacy-preserving and communication-efficient. Then the training data is synthesized based on the aggregated prompts in the server with the foundation diffusion model. The synthetic training data are jointly used to train models, thus can relieve data heterogeneity problem and improve performance. Our framework can implement one-shot FL, and clients do not need to train models locally. We describe the prompt aggregation process in Section 4.1, the synthesis of training data in Section 4.2, one-shot and multi-round model updating in Section 4.3.

## 4.1 PROMPT GENERATION AND AGGREGATION

We investigate two types of prompt generation: class-level prompt and instance-level prompt. The class-level prompts are generated based on class names, providing high-level guidance to the generative model. On the other hand, the instance-level prompt strategy leverages prompts that are tailored for individual instances in the private dataset, which are more informative for training models.

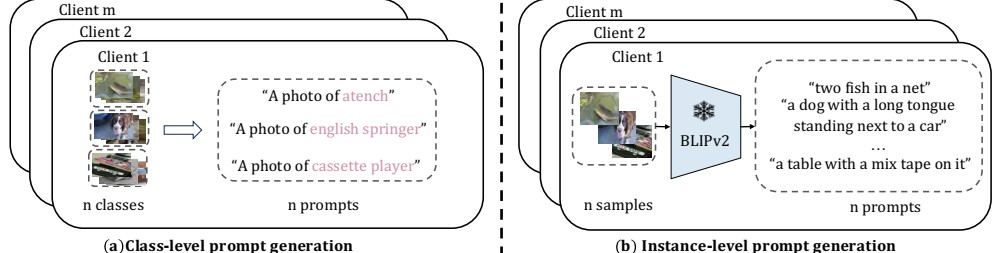

Figure 3: Two ways of prompt generation. The first method requires absolutely no private information from any samples. The second method, on the other hand, relies on textual descriptions of images.

Figure 3 illustrates the two types of prompts. For example, we generate prompt like "A photo of a {class name}" for every class. Each class-level prompt can used to synthesize many images accompanying different noises. We use BLIP-v2 Li et al. (2023c) to generate captions for each real image $x_i$ as the instance-level prompt, denoted as $p_i$. After receiving prompts and corresponding labels $\{(p_i, y_i)\}_{i=1}^{N}$ from all clients, where $N$ is the number of total prompts, the server aggregates them for data synthesis with foundation generative models, i.e. Stable Diffusion.

## 4.2 TRAINING SET SYNTHESIS

After receiving all prompts, the server synthesizes every training sample $s_i$ by prompting the pre-trained Stable Diffusion with each $p_i$ as follows:

$$s_i = G(z_i, p_i) = \sqrt{\beta} \sum_{t=1}^{T} \sqrt{1 - \beta^t} \cdot \frac{1}{\sqrt{T}} \cdot G_{\theta_t}(z_i, p_i), \quad (2)$$

---

**Algorithm 1:** Federated Generative Learning: PyTorch-like Pseudocode

---

```
# for first communication round, at server side.
prompts_list = [], global_model = network()
for client in all_clients:
    # class-level prompts or instance-level prompts
    prompts_client = prompts_generation(client)
    prompts_list.append(prompt)
prompts = prompts_aggregation(prompts_list)
synthetic_dataloader = generative_model(prompts, noises)
# train the global model at server side
server_update(synthetic_dataloader, global_model)

# for 5-round communication
for com_round in [2,3,4,5]:
    local_weights = []
    # randomly seletct m clients
    for client in selected_clients:
        # SGD update on local training data
        weight = local_update(client, global_model)
        local_weights.append(weight)
    # update global model via aggregating local weights
    global_model = model_avegraging(local_weights)
    # finetune global model for highly skewed data distribution at server side.
    if server_finetune: server_update(synthetic_dataloader, global_model)
```

---

**Notes**: The foundation generative model has never seen any private data from any client, and all synthetic data is only generated and stored on the server.

---

where $z_i$ is a random noise vector, $p_i$ is the prompt, and $G_{\theta_t}$ is the denoising network parameterized with $\theta_t$ at time step $t$. The hyperparameter $\beta$ controls the trade-off between image quality and diversity, and $T$ is the number of diffusion steps. The inference process iteratively denoises the image then outputs the final synthetic training image. Finally, the server generates the synthetic training set $\mathcal{S} = \{(s_i, y_i)\}_{i=1}^{N}$. Note that it is easy to synthesize more diverse training samples by combining multiple random noises with the sample prompt, thus training better models. In practice, we can adjust the number of synthetic training samples to trade-off the computational cost and performance. The Pytorch-like pseudocode of our method is presented in Algorithm 1.

### 4.3 MODEL UPDATING

#### 4.3.1 ONE-SHOT UPDATING.

In this paper, we first demonstrate the efficacy of our approach in the context of one-shot federated learning Guha et al. (2019); Zhang et al. (2022a). This involves a central server learning a global model over a network of federated devices in a single round of communication.

After obtaining the synthetic training set $\mathcal{S} = \{(s_i, y_i)\}_{i=1}^{N}$, we jointly train the model in the server, *i.e.*, ResNet18 with cross-entropy loss:

$$\mathcal{L} = -\frac{1}{N} \sum_{i=1}^{N} y_i \log(\hat{y}_i) + (1 - y_i) \log(1 - \hat{y}_i), \tag{3}$$

where $\hat{y}_i$ is the model prediction. Upon completion of global model training, we send the model weights to each client, thereby completing the one-shot communication.

#### 4.3.2 MULTI-ROUND UPDATING

Our method can also implement multi-round communication like traditional FL methods, which can bring further performance improvement. We consider two updating algorithms: updating models *without* and *with* synthetic data in the following rounds of communication. Note that model is trained on the synthetic training set in the first-round communication for both two algorithms.

**Without Synthetic Data.**   After the first round communication, each client receives the updated model from the server and then locally fine-tuning on it on private real training data, *i.e.*, $\theta_k$. After locally fine-tuning, the server collects updated models from all clients for model aggregation, which is formulated as $\theta = \frac{1}{K} \sum_{k=1}^{K} \theta_k$, where $K$ is the total number of clients. In other words, the server only aggregates models after the first round of communication. This process is repeated until the model is converged or reaching the maximum communication rounds $T$.

Table 1: Performance comparison among different methods on subsets of ImageNet. Hyper-parameter $\beta$ controls the degree of label imbalance. We synthesize $20k$ images per class. The improvement $_\uparrow$ is compared to FedAvg IID results.

| Dataset | FedAvg | | | Ours (one-shot) | Ours (5-round) | | | Centralized |
|---|---|---|---|---|---|---|---|---|
| | $\beta = 0.01$ | $\beta = 0.5$ | IID | | $\beta = 0.01$ | $\beta = 0.5$ | IID | |
| ImageNette | 51.6 | 75.0 | 79.2 | $85.2_{\uparrow 6.0}$ | 82.8 | 94.0 | $\mathbf{95.6}_{\uparrow 16.4}$ | 92.2 |
| ImageFruit | 29.0 | 51.2 | 55.6 | $71.8_{\uparrow 16.2}$ | 67.2 | 80.2 | $\mathbf{83.2}_{\uparrow 27.6}$ | 78.2 |
| ImageYellow | 50.6 | 70.2 | 74.6 | $82.4_{\uparrow 7.8}$ | 79.4 | 91.0 | $\mathbf{94.8}_{\uparrow 20.2}$ | 90.8 |
| ImageSquawk | 49.6 | 73.2 | 79.8 | $88.8_{\uparrow 9.0}$ | 90.0 | 95.0 | $\mathbf{95.6}_{\uparrow 15.8}$ | 92.4 |
| ImageNet100 | 36.3 | 44.6 | 49.4 | $48.4_{\downarrow 1.0}$ | 70.1 | 74.9 | $\mathbf{80.1}_{\uparrow 30.7}$ | 77.0 |

**With Synthetic Data.** In the context of few-round communication scenarios, we find that training the aggregated model on synthesized data at the server-side can effectively mitigate the issue of forgetting Lee et al. (2021) after aggregation, albeit at the expense of additional computational overhead. Specifically, fine-tuning the aggregated model on a synthetic training set in each communication round yields a substantial improvement in performance, particularly when dealing with highly imbalanced data distributions among clients. More details are provided in Section 5.2.

## 5 EXPERIMENTAL RESULTS

### 5.1 EXPERIMENTAL SETUPS

**Data Partition**: Follow the setting in Kairouz et al. (2021); Li et al. (2022), we adopt two classic data partitioning strategies, namely, label distribution skew and feature distribution skew:

- **Label Distribution Skew**: Following in Zhang et al. (2022b; 2023b), in which the label distributions varies on different clients, we employ the Dirichlet distribution $p \sim Dir(\beta)$ to simulate imbalanced label distributions. The hyper-parameter $\beta$ controls the degree of label imbalance, where a smaller value of $\beta$ indicates a more skewed label distribution.
- **Feature Distribution Skew**: In this setting, clients share the same label space while different feature distribution, which has been extensively studied in previous work Li et al. (2021); Zhu et al. (2022a); Yao et al. (2022); Gong et al. (2022). We perform the classification task on natural images sourced from DomainNet Peng et al. (2019), which consists of diverse distributions of natural images from six distinct data sources.

**Baselines**: In our experiments, we select the popular FedAvg McMahan et al. (2017) method and centralized training as the baselines. Assume that we have a total of 5 clients and that every client participates in communication. For both centralized training and federated learning, the local learning rate is set to 0.01, and we utilize the SGD optimizer with a momentum of 0.9. During the FedAvg training process, each client performs local updates for 5 epochs, and the communication round is set to 200. The centralized training consists of 120 rounds of iterations.

**Implementation:** We employ the class-level prompt by default, which does not directly utilize local data information, thus providing enhanced privacy protection. We use Stable Diffusion v2-1-base model to construct synthetic data. We evaluate the performance of our method in scenarios of one-round (*i.e.*, **Ours (one-shot)**) and five-round communications (*i.e.*, **Ours (5-round)**). In the one-round communication scenario, we train the model for 120 epochs on the server. In the five-round communication scenario, we implement two variations: 1) **Ours (5-round)**: Based on the model trained in the first round, we perform four additional rounds of communication using the FedAvg algorithm. 2) **Ours (5-round-syn)** : For extreme data distribution skew scenario, we further fine-tune the aggregated model using the synthetic dataset generated on the server during the first round for all five epochs. Please refer to **Appendix** A.1 for more details on implementation and datasets.

### 5.2 RESULTS ON IMAGENET SUBSETS

To evaluate the efficacy of our method on datasets with label distribution skew, we conduct experiments on five subsets of 224×224 ImageNet Russakovsky et al. (2015). Firstly, following Howard; Cazenavette et al. (2022), we do experiments on four datasets with 10 categories each, namely the coarse-grained ImageNette and ImageYellow, and the fine-grained ImageFruit and ImageSquawk. We further conducted experiments on a larger-scale dataset, namely ImageNet100, involving 100 categories. For each dataset, we generate $20k$ images per class using the diffusion model. For the 10-category classification task, we employ the ResNet-18 He et al. (2016) model, while for the

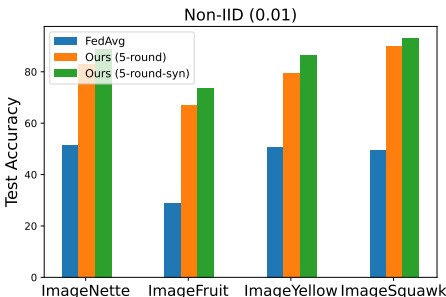 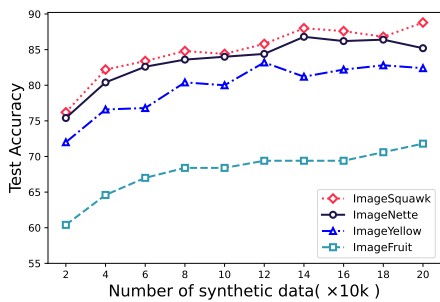

(a) Accuracy on Highly Skewed Data ($\beta = 0.01$).  (b) Performance with Varying Synthetic Data Volume.

Figure 4: **Left**: Performance comparison among FedAvg, Ours (5-round), and Ours (5-round-syn) on four subsets of ImageNet with $\beta = 0.01$. **Right**: Relationship between the number of synthetic images and testing accuracy on four datasets.

100-category classification task, we utilize the ResNet-34 model. We simulate three distinct dataset distributions, *i.e.*, IID, non-IID ($\beta = 0.5$) and highly skewed distribution with $\beta = 0.01$. As the varying data distribution does not influence our method during the initial round of communication for collecting prompts, our results in different settings are the same. Thus, we report a single value.

The overall experimental results are shown in Table 1. It is evident that our method with one-round communication outperforms the FedAvg method with 200 rounds of communication, by 6.0%, 16.2%, 7.8% and 9.0% on the four 10-category datasets in IID setting. Notably, our method is completely insensitive to data distribution in the first round. Hence, under extreme data distribution, *i.e.*, $\beta = 0.01$, our method surpasses FedAvg by 33.6%, 42.8%, 31.8% and 39.2% on the four 10-category datasets respectively. In the case of IID setting, our method, after five rounds of communication, exhibits significant performance improvements over FedAvg, namely 16.4%, 27.6%, 20.2% and 15.8% respectively. It is worth noting that our method with 5 round communication, in IID and $\beta = 0.5$, surpasses the performance of centralized training on all four datasets. On the challenging ImageNet100 dataset, our one-round communication method is slightly (1%) worse than FedAvg with 200 rounds of communication, while outperforms it with label distribution skew by large margins. For 5 round communication, our method exceeds FedAvg by around 30% in different settings.

**Results on Highly Skewed Data.** Table 1 shows that when the data distribution is extremely skewed ($\beta = 0.01$), Ours (5-round) does not outperform Ours (one-shot) on Imagenette, ImageFruit, and ImageYellow datasets. We attribute this phenomenon to the fact that in case of highly skewed data distribution, more rounds of communication are required for the models to converge gradually Zhao et al. (2018); Wang et al. (2020). In order to achieve better results within five rounds, we perform fine-tuning on the aggregated model using the synthesized dataset from the first round on the server for 5 additional epochs before distributing the updated model, namely Ours (5-round-syn). The results are presented in Figure 4(a), which clearly demonstrate that fine-tuning on our synthesized dataset can significantly enhance model performance in scenarios of extreme data distribution.

**Class-level Prompts versus Instance-level Prompts.** In main experiments, we synthesize a number of samples based on class-level prompts, *e.g.*, *A photo of a [class name]*, and random noise using Stable Diffusion. Then following Lei et al. (2023), we caption individual images with foundation models, *e.g.*, BLIP-2 Li et al. (2023c), and synthesize the new image based on the individual caption and random noise. These prompts provide more precise guidance to the generative model by considering the specific characteristics and context of each sample. Note that Zhou et al. (2023) also produce instance-level prompts by diffusion inversion, while the inverted samples will leak data privacy and it is very expensive to implement diffusion inversion on large datasets for clients. In Figure 5, we present the performances of two kinds of prompts, where we synthesize 1300 images per class, the same number as the real training set. It is evident that employing a more precise instance-level prompt leads to higher accuracy, *e.g.*, achieving 78.62% on ImageNette, compared to class-level prompt, which achieves 73.20%. This result clearly highlights the importance of considering more detailed instance-level information in the prompt designing. However, the instance-level prompts involves specific real image information, we show discussions on privacy in Section 5.4.

**Number of Synthetic Data.** By comparing the results of $1.3k$ synthetic images per class in Figure 5 and $20k$ synthetic images per class Table 1, we find the performance of our method can significantly

Figure 5: Accuracy of class-level prompt and instance-level prompt on four datasets. We synthesize 1300 images per class. After using instance-level prompts, the best accuracy improved by 10.2%, 4.6%, 5.8%, and 3.8% respectively.

Table 2: Performance comparison in the scenario of feature distribution skew. Each client hosts data from a specific domain of DomainNet dataset.

| Method | Clipart | Infograph | Painting | Quickdraw | Real | Sketch | Average |
|---|---|---|---|---|---|---|---|
| FedAvg | 80.97 | 41.90 | 57.33 | 78.93 | 80.56 | 70.06 | 72.30 |
| Centralized | 81.37 | 50.82 | 60.63 | 92.46 | 82.20 | 73.93 | 78.10 |
| Ours (one-shot) | $83.40_{\uparrow 2.43}$ | $49.58_{\uparrow 7.68}$ | $76.88_{\uparrow 19.55}$ | $51.80_{\downarrow 27.13}$ | $87.06_{\uparrow 6.5}$ | $81.10_{\uparrow 11.04}$ | $71.59_{\downarrow 0.71}$ |
| **Ours (5-round)** | $\mathbf{90.89}_{\uparrow 9.92}$ | $\mathbf{61.61}_{\uparrow 19.71}$ | $\mathbf{79.52}_{\uparrow 22.19}$ | $\mathbf{81.13}_{\uparrow 2.2}$ | $\mathbf{91.13}_{\uparrow 10.57}$ | $\mathbf{90.20}_{\uparrow 20.14}$ | $\mathbf{84.05}_{\uparrow 11.75}$ |

improve by synthesizing more training samples, *i.e.*, from 73.2% to 85.2% on the Imagenette dataset. We further study the influence of synthetic image number and model performance in Figure 4(b) on four 10-category datasets. We synthesize more images per class by integrating the prompts and more random noises. Obviously, as the number of images per class increases from 2k to 20k, the test accuracy improves consistently for all datasets. Hence, our Federated Generative Learning can easily improve the performance by synthesizing more training samples in the server with little extra cost.

### 5.3 RESULTS ON DOMAINNET SUBSET

To simulate the scenario of feature distribution skew, we select 10 categories from the DomainNet dataset to conduct experiments . Each client is assigned a specific domain, and we have a total of 6 clients participating in FL. To ensure an adequate amount of data, we synthesize 3,500 samples for each class within each domain, resulting in a cumulative dataset of 210$k$ samples. Table 2 presents the performance of various methods on six domains respectively and their average accuracy. It shows that in the one-shot communication scenario, our method outperforms FedAvg by 2% to 19% in five domains, but exhibits notably poor performance in the Quickdraw domain. To investigate the underlying reason, we visualize the synthetic and real data in **Appendix** A.2.1.It becomes apparent that this performance decline is attributed to the difficulty for diffusion model to synthesize images that align with Quickdraw domain when using the class-level prompt, *i.e.*, "A black and white drawing of a {class name}". We leave this problem as the future work. However, when implementing a five-round communication experiment, our method demonstrates a 2.2% performance improvement over FedAvg specifically on the Quickdraw domain, and an overall performance improvement of 11.75%. Interestingly, our method even surpasses the performance of the centralized training models by 5.95%. We provide further results on each domain in **Appendix** A.2.2, and more visualizations on DomainNet and ImageNet can be seen in **Appendix** A.2.3.

### 5.4 PRIVACY ANALYSIS

**Detecting Content Replication and Memorization.** Previous research Carlini et al. (2023); Zhang et al. (2023a); van den Burg & Williams (2021); Somepalli et al. (2023b;a) has indicated that diffusion models store and reproduce specific images from their training dataset during the generation process. Although our training process (with class-level prompts) does not access the private data of clients, we still discuss the potential privacy risks that may arise. Follow the setting in Somepalli et al. (2023a), we conduct image retrieval experiments, which allows us to compare the synthetic images with the original training images and detect any instances of content duplication. We perform a quantitative analysis on 1000 synthetic images across four datasets. For each synthetic image, we search the training set by computing the Cosine similarity

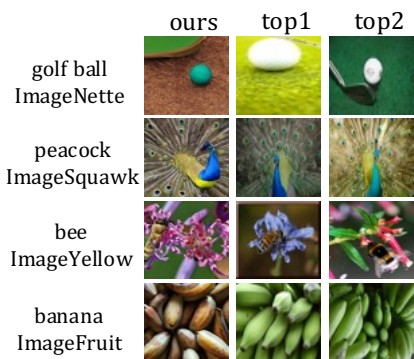

Figure 6: Retrieving similar real training images for each synthetic image.

between its feature and features of real training images. Figure 6 showcases the top 2 most similar images from each of the four datasets. It is obvious that these images exhibit no noteworthy similari-

Table 3: The absolute difference between the average values of member and non-member individuals. The smaller the difference, the better privacy protection.

| Metric | Entropy | | Loss | | Confidence | | Modified Entropy | |
|---|---|---|---|---|---|---|---|---|
| Method | FedAvg | Ours | FedAvg | Ours | FedAvg | Ours | FedAvg | Ours |
| ImageNette | 0.192 | $0.085_{\downarrow 0.107}$ | 0.111 | $0.083_{\downarrow 0.028}$ | 0.119 | $0.087_{\downarrow 0.032}$ | 0.255 | $0.273_{\uparrow 0.018}$ |
| ImageFruit | 0.123 | $0.012_{\downarrow 0.111}$ | 0.100 | $0.020_{\downarrow 0.080}$ | 0.105 | $0.021_{\downarrow 0.084}$ | 0.288 | $0.074_{\downarrow 0.214}$ |
| ImageYellow | 0.073 | $0.051_{\downarrow 0.022}$ | 0.138 | $0.041_{\downarrow 0.097}$ | 0.141 | $0.043_{\downarrow 0.098}$ | 0.529 | $0.110_{\downarrow 0.419}$ |
| ImageSquawk | 0.166 | $0.047_{\downarrow 0.119}$ | 0.113 | $0.042_{\downarrow 0.071}$ | 0.119 | $0.044_{\downarrow 0.075}$ | 0.287 | $0.137_{\downarrow 0.150}$ |

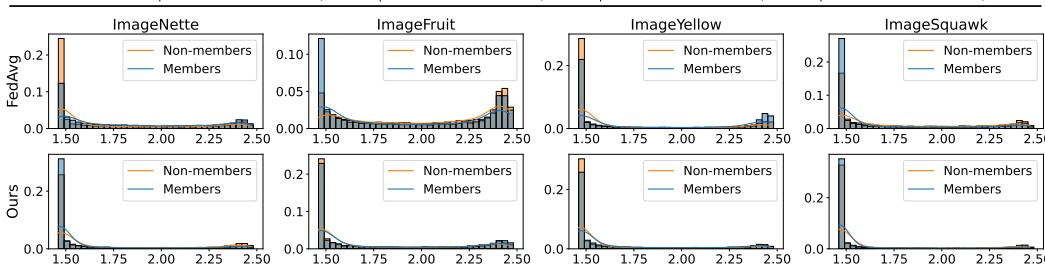

Figure 7: Distribution of Loss metrics between members and non-member. A greater discrepancy in distribution indicates more information leakage from the model regarding its training set.

ties in terms of both background and foreground. This verifies our method does not compromise the privacy of the client's private data.

**Membership Inference Attack (MIA).** The objective of MIA is to examine whether a specific data point belongs to the training set used to train a machine learning model. Due to the fact that our method does not directly use real training data, intuitively it should be able to defend against membership inference attacks. To verify this, we report the results of MIA with the low false-positive rate regime on ImageNette, which is suggested by the state-of-art Likelihood Ratio Attack (LiRA) Carlini et al. (2022). As show in Figure 8, when employing the LiRA against models trained using private data and synthetic data, the

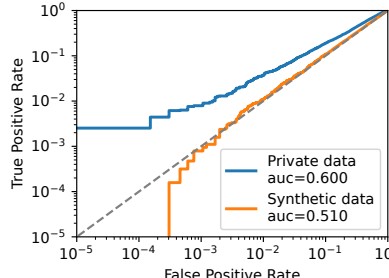

Figure 8: Results under LiRA attack.

latter exhibits a stronger defense against membership inference attacks. This is attributed to the fact that synthetic data hardly contains any sensitive information from the private data.

Also, we employ popular metrics in MIA, such as confidence Salem et al. (2018), loss Sablayrolles et al. (2019); Yeom et al. (2018), entropy Salem et al. (2018), and modified entropy Song & Mittal (2021), to compute the statistical characteristics of member and non-member data under both FedAvg and our method. The greater similarity between the distributions of the model's output metrics (*e.g.*, loss and entropy) for training members and non-members, the more membership privacy the model can guarantee. We calculate the average metric values for training members and non-members, respectively, and then compare the difference between the two averages. The results are presented in Table 3, which demonstrates that our method can reduce the discrepancy between training members and non-members on most metrics compared to FedAvg. In other words, it indicates that our model's output distributions of training members and non-members are more similar. Additionally, we provide the probability distribution of the metrics for training members and non-members, as depicted in Figure 7. Obviously, our model exhibits significantly more similar distributions for training members and non-members compared to FedAvg. The other three metrics can be found in **Appendix** A.2.4.

## 6 CONCLUSION

In this work, we introduce a pioneering framework for Federated Learning, named *Federated Generative Learning*, which transmits prompts associated with distributed training data between clients and the server. By leveraging foundation generative models, informative training data can be synthesized remotely using received prompts that contain minimal privacy. The proposed framework exhibits several noteworthy advantages, including improved communication efficiency, better resilience to distribution shift, substantial performance gains, and enhanced privacy protection. We hope this work can inspire researchers in the field of federated learning to shift from transmitting model parameters/-gradients to the new paradigm of transmitting prompts. Besides, we show detailed discussions on future research in **Appendix** A.3, *e.g.*, non-IID issues, complicated domains, prompts risks.

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

# A APPENDIX

## A.1 EXPERIMENTS SETTING

### A.1.1 PROMPT GENERATION FOR SYNTHETIC DATA IN IMAGENET AND DOMAINNET

In this section, we present the configurations for prompt generation when synthesizing data for the ImageNet and DomainNet datasets. As depicted in Table 4, for class-level prompt generation on ImageNet-like datasets, the prompt template consists of the label followed by ", real-world images, high resolution." Examples of generated prompts include "tench, real world images, high resolution" and "English springer, real world images, high resolution." On DomainNet Subset datasets, the prompt template comprises the label and style, where the label represents the category name, and the style describes the domain. Examples of generated prompts in this context are "an airplane, Sketch drawing with only one object in the picture" and "an airplane, real world images, high resolution, with only one object in the picture."

For instance-level prompt generation on ImageNet Subset datasets, the prompt template consists of the label followed by ", " and the image caption followed by ", real-world images, high resolution." Here, the label represents the category name, and image caption corresponds to the textual description of the image generated by BLIPv2. Examples of generated prompts include "tench, Tinca tinca, a man kneeling down holding a fish in the grass" and "tench, Tinca tinca, a man kneeling down holding a large fish in the water."

Table 4: Examples of prompt generation patterns for Class-Level and Instance-Level prompting

| Prompt Type | Dataset | Prompt Template | Prompt Example |
|---|---|---|---|
| Class Level | ImageNet Subset | label + ', real world images , high resolution' | tench, real world images, high resolution |
| Class Level | DomainNet Subset | label + style | an airplane, Sketch drawing with only one object in the picture |
| Instance Level | ImageNet Subset | label + ', ' + image caption + ', real world images, high resolution.' | tench, Tinca tinca, a man kneeling down holding a fish in the grass |

### A.1.2 DATASET DESCRIPTION

In this section, we provide a detailed description of the datasets used in our experiments. The datasets include the DomainNet Subset, lmageNette, lmageFruit, lmageYellow, lmageSquawk, and ImageNet100.

**DomainNet Subset**: This subset is selected from the DomainNet dataset and consists of ten categories spanning six different domains. Refer to Table 5 for detailed class and domain names.

Table 5: Detailed description of the DomainNet Subset Dataset

| Description | # class | Class name | Domain name |
|---|---|---|---|
| 10 classes from DomainNet | 10 | airplane, clock, axe, basketball, bicycle, bird, strawberry, flower, pizza, bracelet | Clipart, Infograph, Painting, Quickdraw, Real, Sketch |

**ImageNet Subset**: These datasets are subsets extracted from ImageNet, including lmageNette, lmageFruit, lmageYellow, lmageSquawk, and ImageNet100. Refer to Table 6 for details on class names and class IDs.

## A.2 ADDITIONAL EXPERIMENTS

### A.2.1 SYNTHETIC DATA VISUALIZATION ON QUICKDRAW DOMAIN

To investigate the reasons behind the notably poor performance in the Quickdraw domain, we visualize both synthetic and real data for the Painting and QuickDraw domains of the DomainNet

Table 6: Detailed description of the ImageNet Subset Dataset

| Dataset | Description | # Class | Class name | Class id |
|---|---|---|---|---|
| lmageNette | 10 class from ImageNet | 10 | (tench, English springer, cassetteplayer, chain saw, church, Frenchhorn, garbage truck, gas pump, golfball, parachute) | (0, 217, 482, 491, 497, 566, 569,571, 574, 701) |
| lmageFruit | 10 class from ImageNet | 10 | (pineapple, banana, strawberry, orange, lemon, pomegranate, fig, bell pepper, cucumber, green apple) | (953, 954, 949, 950, 951, 957, 952, 945, 943, 948) |
| lmageYellow | 10 class from ImageNet | 10 | (bee, ladys slipper, banana, lemon, corn, school bus, honeycomb, lion, garden spider, goldfinch) | (309,986, 954, 951, 987, 779, 599, 291, 72, 11) |
| lmageSquawk | 10 class from ImageNet | 10 | (peacock, flamingo, macaw, pelican, king penguin, bald eagle, toucan, ostrich, black swan, cockatoo) | (84, 130, 88, 144, 145, 22, 96, 9, 100, 89) |
| ImageNet100 | 100 class from ImageNet | 100 | - - | - - |

dataset in Figure 9. The first and second rows correspond to real data and synthetic data for the Painting Domain, respectively. It is evident that the synthetic data closely resembles the real data in style, which results in the model performing well in this domain. The third and fourth rows represent real data and synthetic data for the Quickdraw Domain, respectively. Notably, there is a substantial stylistic difference between the synthetic data and real data in this domain, providing an explanation for the poor model performance.

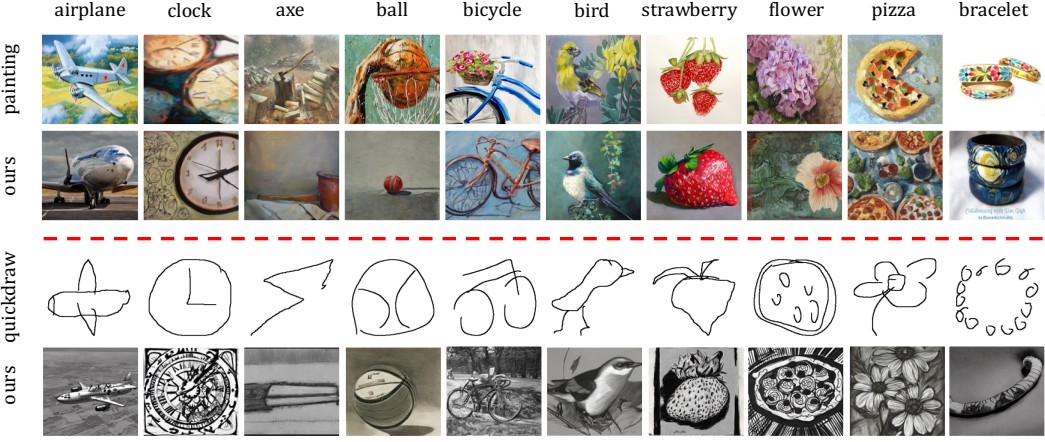

Figure 9: Visualization of original and synthetic data on the Painting and QuickDraw domains of DomainNet dataset. Obviously, it is easy for diffusion model to synthesize painting-like images, while challenging to synthesize images with QuickDraw style.

### A.2.2 TESTING ACCURACY ON EACH DOMAIN.

In this section, we compare the performance of the "Centralized" and "Ours (One-shot)" methods on the DomainNet dataset across different domains during the training process. The experimental

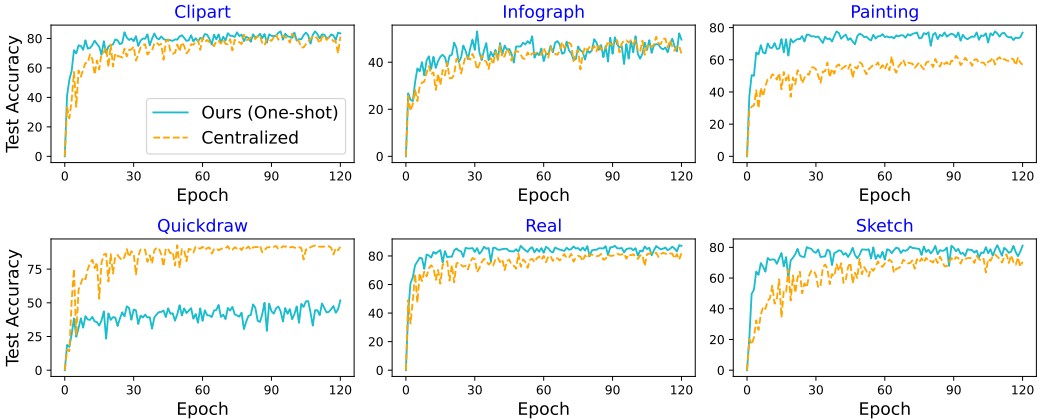

Figure 10: Performance comparison between Ours (One-shot) and the Centralized Training with increasing number of training epochs on six domains of DomainNet dataset.

results, as shown in Figure 10, demonstrate that, except for the "Quickdraw" domain, Ours (One-shot) outperforms the Centralized Training in five domains. Particularly, in the "Painting" domain, our method achieves a significant performance improvement. We provide further explanation about the performance gap in "Quickdraw" domain through visualization in the following section.

### A.2.3 VISUALIZATION ON DOMAINNET AND IMAGENET

In this section, we provide comprehensive visualizations of the synthetic data generated from the ImageNet and DomainNet datasets. Figure 11 showcases visualizations of synthetic data from four distinct subsets of the ImageNet dataset. Each pair of rows corresponds to one of these subsets, which includes ImageNette, ImageFruit, ImageYellow, and ImageSquawk subdatasets. Within each column, individual images represent specific classes from these subsets. Figure 12 offers a glimpse into the synthetic data generated for six domains within the DomainNet dataset. Similar to the ImageNet visualization, each pair of rows represents one of these domains, which encompasses sketch, real, quickdraw, painting, infograph, and clippart domains. Within each column, you will find synthetic images representing individual classes within the respective domain. Upon close examination, it becomes readily apparent that the synthetic data demonstrates striking color accuracy, precise delineation of object boundaries, and an impressive level of realism that closely approximates that of genuine real-world images. The synthetic images not only maintain vivid and faithful color representations but also capture intricate details, ensuring that the synthetic data closely mirrors the characteristics found in authentic visual data.

### A.2.4 MIA RESULTS ON 3 METRICS

In this section, we compare the distributions of three metrics, namely, the classifier's confidence, the classifier's entropy, and the classifier's modified entropy, between training members and non-members. The distribution of the classifier's confidence is depicted in Figure 13, the distribution of the classifier's entropy is illustrated in Figure 14, and the distribution of the classifier's modified entropy is shown in Figure 15. Notably, it can be clearly observed that, in comparison to the FedAvg method, our model exhibits significantly more similar distributions between training members and non-members in these three metrics.

### A.3 DISCUSSION ON FUTURE DIRECTIONS

This paper has discussed the advantages of using generative models for synthetic training data, which include preserving privacy and achieving high-performance models. There are still challenges to be addressed in the field of Federated Generative Learning, which we leave as potential areas of future research:

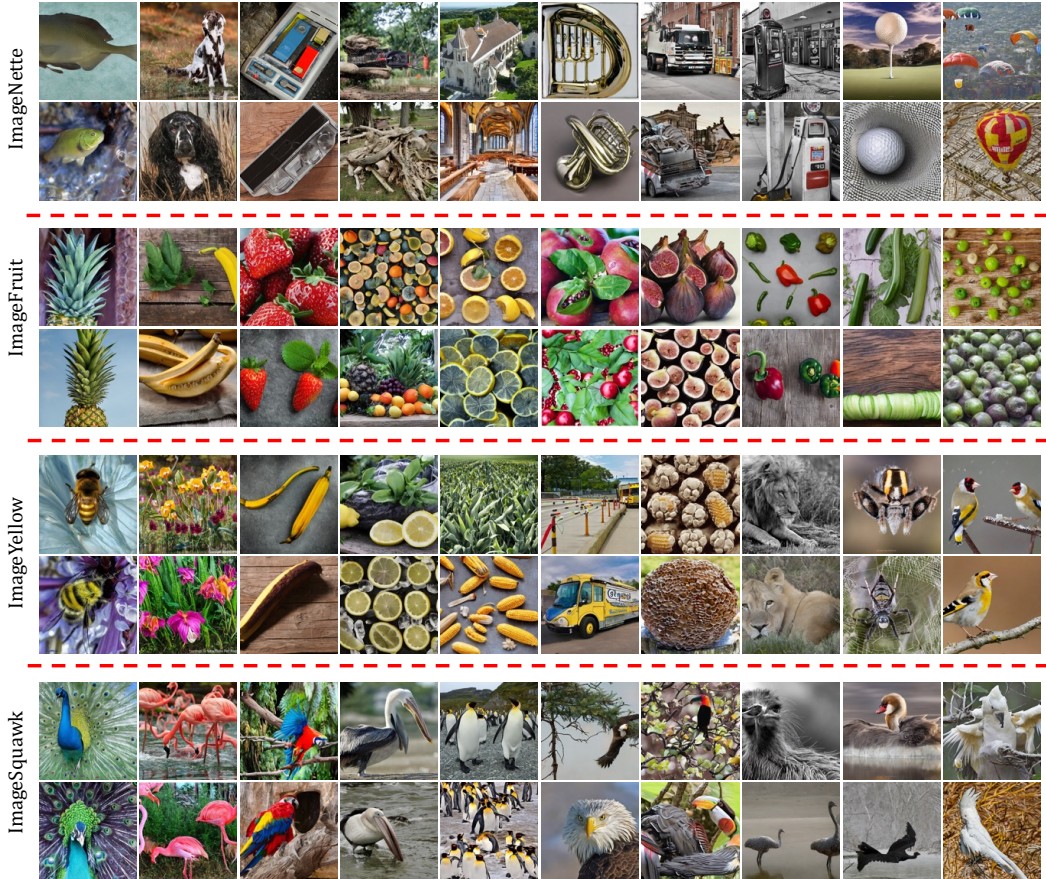

Figure 11: Visualization of synthetic images of four ImageNet subsets.

**The Non-IID Issue**  One fundamental challenge in federated learning is the non-IID (non-Independently and Identically Distributed) nature of data across clients, resulting from varying data distributions among different clients. Traditional approaches to addressing non-IID issues often focus on optimization techniques. However, these methods do not fundamentally alleviate the problem of model accuracy degradation caused by non-IID data. Indeed, by leveraging generative models, we have the potential to mitigate this issue by filling in missing data on individual clients, aligning each client's data distribution with the global data distribution. In such cases, the federated learning training will closely resemble centralized training.

**Enhancing Synthesis Capabilities in Complex Domains**  In the realm of Federated Generative Learning, a key challenge lies in augmenting the synthesis capabilities, particularly when dealing with complex domains. While ImageNet features closely resemble real-world image characteristics, there may exist an overlap between the pre-training dataset of the diffusion model, LAION-5B Schuhmann et al. (2022b), and ImageNet, enabling the generation of high-quality images. However, in domains such as medical imaging and remote sensing, the generalization performance of the diffusion model may be insufficient.

A viable strategy to bolster its performance in specific domains involves the rapid fine-tuning of the diffusion model using techniques like LoRA Hu et al. (2021). However, this approach raises a significant concern: does the fine-tuning of the diffusion model compromise the privacy of sensitive private data? Specifically, when fine-tuning is performed using data from a particular domain, there is a risk that the model could inadvertently memorize or expose private information contained within the training data. Future research should focus on developing techniques to enhance domain-specific performance while rigorously addressing privacy implications.

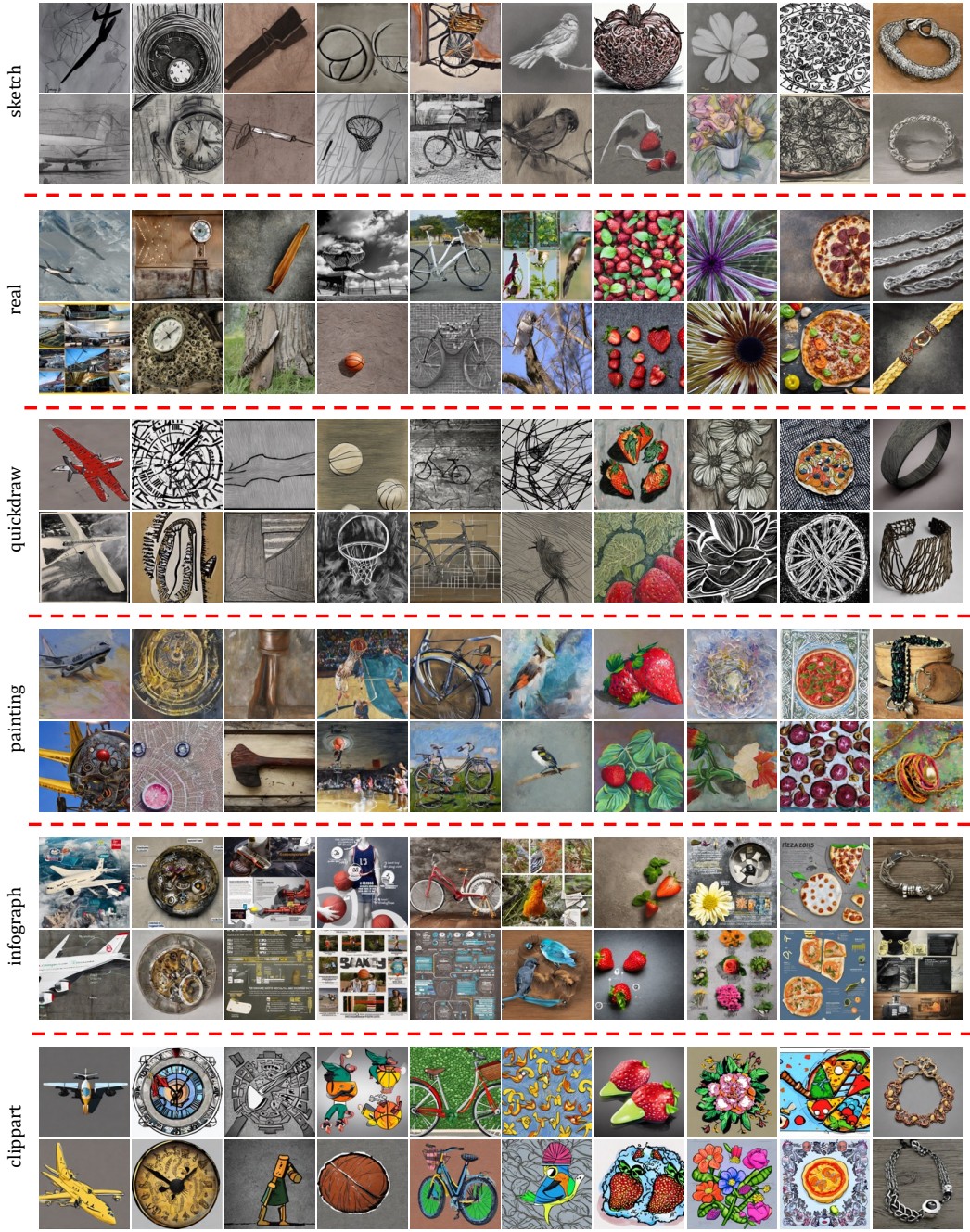

Figure 12: Visualization of synthetic images of six domains from DomainNet dataset.

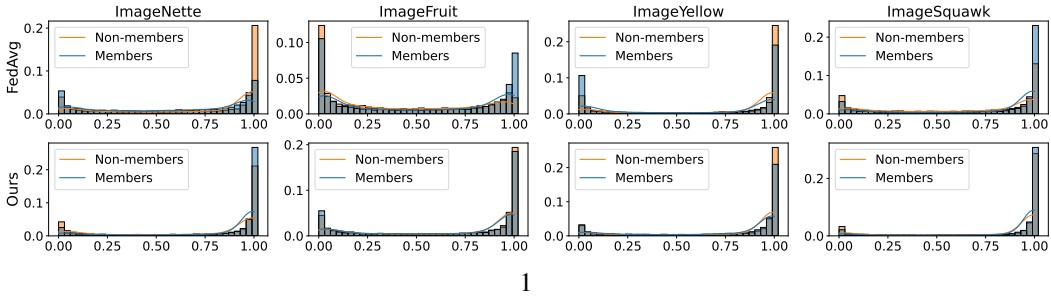

Figure 13: Distribution of the classifier's **confidence** between members and non-member samples. A greater disparity in distribution indicates a higher degree of information leakage from the model regarding its training set.

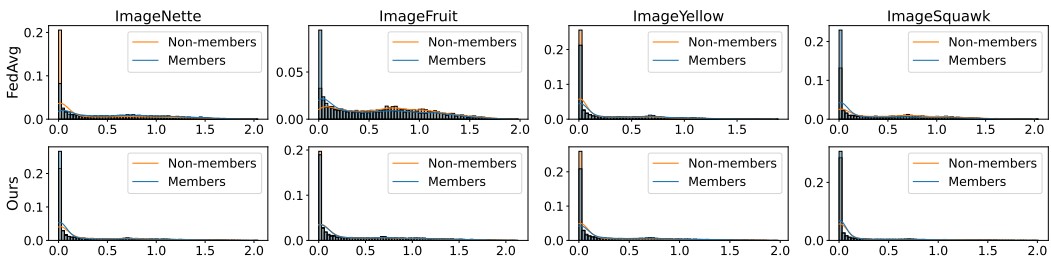

Figure 14: Distribution of the classifier's **entropy** between members and non-member samples. A greater disparity in distribution indicates a higher degree of information leakage from the model regarding its training set.

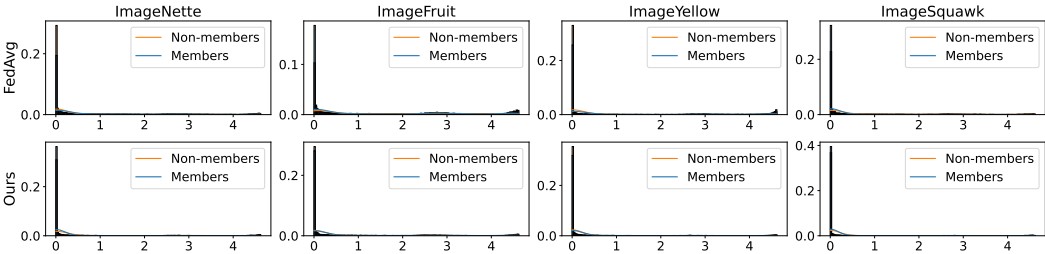

Figure 15: Distribution of the classifier's **modified entropy** between members and non-member samples. A greater disparity in distribution indicates a higher degree of information leakage from the model regarding its training set.

**Risks Associated with Prompts**    During the communication phase, traditional Federated Learning methods typically transmit model parameters. If intercepted by an adversary, these parameters can be exploited for adversarial attacks and model inversion attacks. While using prompts for communication enhances security, the potential for attackers to reconstruct private data using prompts remains a topic with limited research, both in black-box and white-box scenarios.

In conclusion, Federated Generative Learning offers promising avenues for enhancing privacy-preserving machine learning while achieving high-performance models. Addressing the challenges of resolving non-IID issues, synthesizing data in complex domains, and mitigating the risks associated with prompts is vital for the advancement of this field. Future research should aim to tackle these challenges to realize the full potential of Federated Generative Learning.

### A.4    MORE EXPERIMENTS

#### A.4.1    RESULTS ON MORE CHALLENGING DATASETS

Even for particularly challenging domains such as remote sensing images or fine-grained classification datasets, our method can easily adapt to these scenarios. We conducted experiments on several fine-grained image classification datasets, namely CUB-200 Wah et al. (2011), Stanford Cars Krause et al. (2013), and also the satellite image dataset EuroSAT Helber et al. (2019). CUB-200 is a challenging dataset consisting of 200 bird species, while Stanford Cars contains 16,185 images belonging to 196 classes of cars. The size of fine-grained recognition datasets is typically smaller compared to general image classification datasets. In previous work Zhu et al. (2022b); Diao et al. (2022), a common practice is to utilize a pretrained model that has been trained on the ImageNet dataset. In this study, we present two approaches: training the model from scratch and loading a pretrained ResNet34 model. As shown in Table 7, our method achieves excellent performance even in these challenging domains. Additionally, in the cross-silo federated learning scenario, when clients have strong computational capabilities, one can simply finetune the foundation models on these domains, achieving better performance than normal federated learning methods.

Table 7: Results on three challenging datasets.

| Prompt type | Settings Training type | Dataset | FedAvg $\beta = 0.01$ | $\beta = 0.5$ | FedAvg (IID) | Ours (one-shot) | Ours (5-round) $\beta = 0.01$ | $\beta = 0.5$ | IID | Centralized |
|---|---|---|---|---|---|---|---|---|---|---|
| instance | scratch | CUB-200 | 35.04 | 36.61 | 36.62 | 44.17 | 64.53 | 69.19 | 71.01 | 48.31 |
| instance | pretrain | CUB-200 | 78.98 | 79.08 | 78.48 | 54.02 | 75.13 | 78.96 | 80.72 | 81.77 |
| class | scratch | CUB-200 | 35.04 | 36.61 | 36.62 | 45.34 | 67.66 | 71.9 | 73.33 | 48.32 |
| class | pretrain | CUB-200 | 78.98 | 79.08 | 78.48 | 52.73 | 74.68 | 78.7 | 80.32 | 81.77 |
| class | scratch | Cars | 55.18 | 42.43 | 44.48 | 54.48 | 83.31 | 87.22 | 88.07 | 64.72 |
| class | pretrain | Cars | 87.71 | 88.91 | 88.96 | 60.55 | 87.31 | 90.05 | 90.73 | 91.21 |
| class | scratch | EuroSAT | 43.94 | 74.48 | 84.87 | 38.37 | 37.59 | 82.94 | 91.01 | 94.31 |

#### A.4.2    RESULTS ON MORE CLIENTS

To demonstrate the scalability of our method to a larger number of clients, we extended our analysis to include the results obtained from the ImageNette dataset with 50 and 100 clients. As depicted in Table 8, our method continues to exhibit superior performance compared to FedAvg across all scenarios. Additionally, the improvements achieved by our method remain significant.

Table 8: Results on more clients under non-IID ($\beta = 0.5$).

| Method | FedAvg | FedOpt | Moon | Ours (one-shot) | Ours (5-round) |
|---|---|---|---|---|---|
| ImageNette | 72.01 | 73.21 | 74.27 | 85.21 | 93.80 |
| ImageNet100 | 40.13 | 41.25 | 41.43 | 48.31 | 72.67 |

#### A.4.3    VARYING THE FOUNDATION MODELS

To investigate the impact of various generative models on the results, we followed the setting in Li et al. (2023d). Our experiments primarily focus on three prevalent conditional diffusion models:

Table 9: Ablation study on the generative model used in FGL.

| Method | one-shot | 5-round, $\beta = 0.01$ | 5-round, $\beta = 0.5$ | IID | Centralized |
|--------|----------|--------------------------|-------------------------|-----|-------------|
| Ours w/ SD | 85.2 | 82.8 | 94.1 | 95.6 | 92.2 |
| Ours w/ Glide | 79.0 | 76.2 | 89.4 | 89.4 | 92.2 |
| Ours w/ Dit | 76.2 | 74.6 | 90.2 | 92.8 | 92.2 |
| FedAvg (120-round) | - | 51.6 | 75.1 | 79.2 | 92.2 |

DiT Peebles & Xie (2023), GLIDE Nichol et al. (2022), and Stable Diffusion. We use these off-the-shelf models to generate synthetic images. Specifically, for GLIDE and Stable Diffusion, the prompt was configured as "a photo of {label name}, real-world images, high resolution". For DiT, the input comprised the label ID corresponding to the ImageNet1k dataset. The images synthesized by DiT and GLIDE are of dimensions 256x256, whereas those produced by Stable Diffusion are of dimensions 512x512. As shown in Table 9, even when we vary the foundation models used in our method, FGL consistently outperforms FedAvg by a significant margin. This observation serves as evidence for the generality of our approach.

