# OpenReview forum: "Federated Generative Learning with Foundation Models"
_ICLR.cc/2024/Conference — Submitted to ICLR 2024_

### Official Review · Reviewer_zu8q · 2023-10-29

**Soundness:** 2 fair
**Presentation:** 3 good
**Contribution:** 2 fair
**Rating:** 5
**Confidence:** 4

**Summary:**

The Federated Generative Learning (FGL) framework offers a novel approach to federated learning, leveraging foundational generative models like Stable Diffusion to generate training data from prompts shared by clients. Clients contribute class-level or instance-level prompts, encapsulating key features of their local data. The server, in turn, amalgamates these prompts and synthesizes corresponding training data for global model training. This approach trims down communication costs since only concise prompts, and not bulky gradients or models, are transferred. This system also boasts robustness to data diversity and has demonstrated superior performance – with just one communication round, it outdid FedAvg's 200 rounds in accuracy. When trialed on skewed ImageNet100 distributions, FGL exceeded FedAvg's performance by 30% in just five communication rounds. Apart from being efficient, FGL also enhances privacy, as prompts reveal lesser private data than traditional methods. Evaluations confirmed no private data memorization in the synthetic images and an enhanced resilience against membership inference attacks. However, challenges persist with non-IID data, intricate domains, and the potential risks associated with prompts.

**Strengths:**

1.	Novel idea of using foundation models to synthesize training data for federated learning, enabling low communication costs and better privacy.
2.	Compelling experimental results demonstrating accuracy improvements over traditional FedAvg, especially with skewed data distributions.
3.	Thorough analysis and quantification of privacy benefits, showing reduced memorization and vulnerability to membership inference attacks.

**Weaknesses:**

1.	The evaluation of the Federated Generative Learning (FGL) framework is limited to simpler domains like ImageNet and doesn't extend to other areas, casting doubt on whether prompts can encapsulate complexity.
2.	While FGL aids in data generation for non-IID data, achieving congruence with a global distribution is yet to be addressed.
3.	Security risks of prompts require more analysis. Could prompts be reverse-engineered to obtain private data?
4.	The framework hasn't been benchmarked against other federated learning methods that employ generative models.

**Questions:**

please refer to the weakness

---

> ### Author Response · Authors · 2023-11-17
> **Response to Reviewer zu8q [1/2]**
>
> Thank you for your constructive comments. We hope the following clarifications can address your concerns.
>
> **Q1: The evaluation of the Federated Generative Learning (FGL) framework is limited to simpler domains like ImageNet and doesn't extend to other areas, casting doubt on whether prompts can encapsulate complexity.**
>
> **A1**: To further validate the effectiveness of our method, we conducted experiments on several fine-grained image classification datasets, namely CUB-200, Stanford Cars, and also the satellite image dataset EuroSAT. CUB-200 is a challenging dataset consisting of 200 bird species, while Stanford Cars contains 16,185 images belonging to 196 classes of cars. As for EuroSAT, the official dataset did not provide predefined training and testing splits, so we performed a split in an 8:2 ratio. The size of fine-grained recognition datasets is typically smaller compared to general image classification datasets. In previous work, a common practice is to utilize a pretrained model that has been trained on the ImageNet dataset. In this study, we present two approaches: training the model from scratch and loading a pretrained ResNet34 model.
> As shown in the table, our method achieves excellent performance even in these challenging domains. This can be attributed to the fact that regardless of the magnitude of domain differences, pretraining a well-performing model on our synthetic data is beneficial for the downstream federated tasks. We have added these results in the appendix.
>
> | |               |         |       |             |              ||  |             |       |  |
> |:---------------:|:-------------:|:-------:|:------------:|:-----------:|:------------:|:---------------:|:--------------:|:-----------:|:-----:|:-----------:|
> |   Prompt type   | Training type | Dataset | FedAvg,$\beta=0.01$ |FedAvg, $\beta=0.5$ | FedAvg (IID) |    Ours (one-shot)              |  Ours (5-round) ,$\beta=0.01$  |Ours (5-round),$\beta=0.5$ |  Ours (5-round),IID  |        Centralized     |
> |     instance    |    scratch    | CUB-200 |     35.04    |    36.61    |     36.62    |      44.17      |      64.53     |    69.19    | 71.01 |    48.31    |
> |     instance    |   pretrain    | CUB-200 |     78.98    |    79.08    |     78.48    |      54.02      |      75.13     |    78.96    | 80.72 |    81.77    |
> |      class      |    scratch    | CUB-200 |     35.04    |    36.61    |     36.62    |      45.34      |      67.66     |     71.9    | 73.33 |    48.32    |
> |      class      |   pretrain    | CUB-200 |     78.98    |    79.08    |     78.48    |      52.73      |      74.68     |     78.7    | 80.32 |    81.77    |
> |      class      |    scratch    |   Cars  |     55.18    |    42.43    |     44.48    |      54.48      |      83.31     |    87.22    | 88.07 |    64.72    |
> |      class      |   pretrain    |   Cars  |     87.71    |    88.91    |     88.96    |      60.55      |      87.31     |    90.05    | 90.73 |    91.21    |
> |      class      |    scratch    | EuroSAT |     43.94    |    74.48    |     84.87    |      38.37      |      37.59     |    82.94    | 91.01 |    94.31    |
>
>
>
> **Q2: While FGL aids in data generation for non-IID data, achieving congruence with a global distribution is yet to be addressed.**
>
> **A2**: Thank you for your valuable feedback. We acknowledge that FGL is effective in generating data for non-IID scenarios, aligning it with a global distribution (e.g., IID settings) also works in our experiments (see Table 1 for IID results).
>
>
> **Q3: Security risks of prompts require more analysis. Could prompts be reverse-engineered to obtain private data?**
>
> **A3**:  During the communication phase, traditional Federated Learning (FL) methods typically transmit model parameters or gradients. However, these parameters can be vulnerable to adversarial attacks and model inversion attacks if intercepted by an adversary. To enhance security, some FL methods utilize prompts for communication. However, the potential for attackers to reconstruct private data using prompts has received limited research attention, both in black-box and white-box scenarios.
> Recent work[1,2] has identified risks associated with the reconstruction of pretrained data in Diffusion models. Nevertheless, there is currently no available method that can solely reconstruct previously unseen private data in Diffusion models based solely on prompts. This presents an interesting and promising research direction for future investigations.
> Consequently, considering the lack of research in this area, our method can be regarded as relatively safe and privacy-preserving.
>
> [1]Shen, Xinyue, et al. "Prompt Stealing Attacks Against Text-to-Image Generation Models." arXiv preprint arXiv:2302.09923 (2023).
>
> [2]Carlini, Nicolas, et al. "Extracting training data from diffusion models." 32nd USENIX Security Symposium (USENIX Security 23). 2023.

---

> ### Author Response · Authors · 2023-11-17
> **Response to Reviewer zu8q [2/2]**
>
> **Q4: The framework hasn't been benchmarked against other federated learning methods that employ generative models.**
>
> **A4**: Unfortunately, we were unable to find any existing methods in the literature that directly address our specific setting, making it difficult to perform a fair comparison. In light of this, and following the suggestions of other reviewers, we conducted experiments using various types of generative models to demonstrate the applicability of our proposed method. To investigate the impact of various generative models on the results, we followed the setting in [1]. Our experiments primarily focus on three prevalent conditional diffusion models: DiT[2], GLIDE[3], and Stable Diffusion. We use these off-the-shelf models to generate synthetic images. Specifically, for GLIDE and Stable Diffusion, the prompt was configured as "a photo of {label name}, real-world images, high resolution." For DiT, the input comprised the label ID corresponding to the ImageNet1k dataset. The images synthesized by DiT and GLIDE are of dimensions 256x256, whereas those produced by Stable Diffusion are of dimensions 512x512. As shown in the following table, even when we vary the foundation models used in our method, FGL consistently outperforms FedAvg by a significant margin. This observation serves as evidence for the generality of our approach. We have added these results in the appendix.
>
> |           Method          | one-shot | 5-round, beta=0.01 | 5-round, beta=0.5 |    IID   |
> |:-------------------------:|:--------:|:------------------:|:-----------------:|:--------:|
> | Ours w/  Stable Diffusion | **85.2** |      **82.8**      |       **94.1**      | **95.6** |
> |       Ours w/ Glide       |    79.0    |        76.2        |        89.4       |   89.4   |
> |        Ours w/ Dit        |   76.2   |        74.6        |        90.2       |   92.8   |
> |     FedAvg (120-round)    |     -    |        51.6        |        75.1       |   79.2   |
>
> [1] Li, Zheng, et al. "Is Synthetic Data From Diffusion Models Ready for Knowledge Distillation?." arXiv preprint arXiv:2305.12954 (2023).
>
> [2] Nichol, Alex, et al. "Glide: Towards photorealistic image generation and editing with text-guided diffusion models." arXiv preprint arXiv:2112.10741 (2021).
>
> [3]Peebles, William, and Saining Xie. "Scalable diffusion models with transformers." Proceedings of the IEEE/CVF International Conference on Computer Vision. 2023.

---

> ### Author Response · Authors · 2023-11-20
> **A friendly reminder that the discussion stage will be closed in 2 days**
>
> Dear Reviewer,
>
> Thank you once again for your valuable comments. As the discussion stage is coming to a close in 2 days, we kindly request your feedback on whether our response adequately addresses your concerns. We would greatly appreciate any additional feedback you may have.
>
> Thank you in advance!
>
> Kind regards,
>
> Authors

---

> ### Author Response · Authors · 2023-11-22
> **With the hope that our response addresses your concerns**
>
> Dear Reviewer zu8q,
>
> As the discussion period is closing, we sincerely look forward to your feedback. The authors deeply appreciate your valuable time and efforts spent reviewing this paper and helping us improve it.
>
> Please also let us know if there are further questions or comments about this paper. We strive to improve the paper consistently, and it is our pleasure to have your feedback!
>
> Best regards,
>
> Authors

---

### Official Review · Reviewer_ZNb2 · 2023-10-29

**Soundness:** 2 fair
**Presentation:** 2 fair
**Contribution:** 2 fair
**Rating:** 5
**Confidence:** 4

**Summary:**

The paper addresses efficiency and client-shift issues in federated learning by harnessing generative foundation models. Unlike traditional approaches that communicate model parameters, this work exploits clients to send instance-level or class-level prompts, generated by a pre-trained captioning model, to the server. The server aggregates these prompts to produce a proxy dataset via a pre-trained generative model, enabling standard federated learning on this dataset. The server then dispatches the refined weights back to the clients. Empirical evaluations underscore the efficacy of the proposed approach.

**Strengths:**

1. The proposed approach significantly reduces communication costs compared to traditional parameter transmission.
2. By leveraging foundation models to synthesize proxy data, the authors effectively mitigate the client-shift problem.
3. A variety of experimental settings across four datasets demonstrate the robustness and effectiveness of the proposed method.

**Weaknesses:**

1. The training framework is predominantly tailored for image datasets, limiting its applicability.
2. The method heavily depends on the congruence between the captioning and generative models, making it challenging to ensure the proxy dataset's distribution aligns with the private data.
3. The experimental setup, with only five clients, may not adequately represent real-world scenarios; expanding the evaluation to include 50 or 100 clients could provide more insightful results.
4. The comparison to a single baseline, FedAvg, falls short; including comparisons to advanced Federated Learning frameworks could better highlight the proposed method's effectiveness.
5. Table 2 shows the proposed method outperforming centralized learning significantly; a thorough explanation of this phenomenon is warranted.

**Questions:**

1. I wonder if the approach cam be applied to other types of datasets, besides the image datasets.
2. What the experimental results would be when the number of clients becomes bigger, e.g., 100.

---

> ### Author Response · Authors · 2023-11-17
> **Response to Reviewer ZNb2 [1/2]**
>
> Thank you for your valuable time in reviewing our paper. Below are responses to your concerns. Please let us know if you require any further information, or if anything is unclear.
>
> **Q1: The training framework is predominantly tailored for image datasets, limiting its applicability.**
>
> **A1**: Our approach is based on existing generative models that are widely used in various domains, such as Computer Vision with Stable Diffusion and Natural Language Processing with GPTs. This means that our framework can easily be applied to other domains, including NLP. However, due to time constraints, we were unable to conduct additional experiments on NLP tasks. We believe that further research in this area would be valuable and should be pursued in the future.
>
> **Q2: The method heavily depends on the congruence between the captioning and generative models, making it challenging to ensure the proxy dataset's distribution aligns with the private data.**
>
> **A2**: To further validate the effectiveness of our method, we conducted experiments on several fine-grained image classification datasets, namely CUB-200, Stanford Cars, and also the satellite image dataset EuroSAT. CUB-200 is a challenging dataset consisting of 200 bird species, while Stanford Cars contains 16,185 images belonging to 196 classes of cars. As for EuroSAT, the official dataset did not provide predefined training and testing splits, so we performed a split in an 8:2 ratio. The size of fine-grained recognition datasets is typically smaller compared to general image classification datasets. In previous work, a common practice is to utilize a pretrained model that has been trained on the ImageNet dataset. In this study, we present two approaches: training the model from scratch and loading a pretrained ResNet34 model.
> As shown in the table, our method achieves excellent performance even in these challenging domains. This can be attributed to the fact that regardless of the magnitude of domain differences, pretraining a well-performing model on our synthetic data is beneficial for the downstream federated tasks.
>
> | |               |         |       |             |              ||  |             |       |  |
> |:---------------:|:-------------:|:-------:|:------------:|:-----------:|:------------:|:---------------:|:--------------:|:-----------:|:-----:|:-----------:|
> |   Prompt type   | Training type | Dataset | FedAvg,$\beta=0.01$ |FedAvg, $\beta=0.5$ | FedAvg (IID) |    Ours (one-shot)              |  Ours (5-round) ,$\beta=0.01$  |Ours (5-round),$\beta=0.5$ |  Ours (5-round),IID  |        Centralized     |
> |     instance    |    scratch    | CUB-200 |     35.04    |    36.61    |     36.62    |      44.17      |      64.53     |    69.19    | 71.01 |    48.31    |
> |     instance    |   pretrain    | CUB-200 |     78.98    |    79.08    |     78.48    |      54.02      |      75.13     |    78.96    | 80.72 |    81.77    |
> |      class      |    scratch    | CUB-200 |     35.04    |    36.61    |     36.62    |      45.34      |      67.66     |     71.9    | 73.33 |    48.32    |
> |      class      |   pretrain    | CUB-200 |     78.98    |    79.08    |     78.48    |      52.73      |      74.68     |     78.7    | 80.32 |    81.77    |
> |      class      |    scratch    |   Cars  |     55.18    |    42.43    |     44.48    |      54.48      |      83.31     |    87.22    | 88.07 |    64.72    |
> |      class      |   pretrain    |   Cars  |     87.71    |    88.91    |     88.96    |      60.55      |      87.31     |    90.05    | 90.73 |    91.21    |
> |      class      |    scratch    | EuroSAT |     43.94    |    74.48    |     84.87    |      38.37      |      37.59     |    82.94    | 91.01 |    94.31    |
>
> **Q3: The experimental setup, with only five clients, may not adequately represent real-world scenarios; expanding the evaluation to include 50 or 100 clients could provide more insightful results.**
>
> **A3**:Thanks for your suggestion. We extended our analysis to include the results obtained from the ImageNette dataset with 50 and 100 clients. As depicted in the table, our method continues to exhibit superior performance compared to FedAvg across all scenarios. Additionally, the improvements achieved by our method remain significant. See more details in the Appendix.
>
> | # Client | FedAvg, $\beta$=0.5 | FedAvg, IID | Ours (one-shot) | Ours (5-round), $\beta$=0.5 | Ours (5-round), IID | Centralized |
> |:--------:|:----------------:|:-----------:|-----------------|:------------------------:|:-------------------:|-------------|
> |     5    |        75.0        |     79.2    |       85.2      |            94.0            |         95.6        |     92.2    |
> |    50    |        72.1        |      77.0     |       85.2      |           93.8           |         91.2        |     92.2    |
> |    100   |        70.1        |     67.2    |       85.2      |           92.8           |         93.2        |     92.2    |

---

> ### Author Response · Authors · 2023-11-17
> **Response to Reviewer ZNb2 [2/2]**
>
> **Q4: The comparison to a single baseline, FedAvg, falls short; including comparisons to advanced Federated Learning frameworks could better highlight the proposed method's effectiveness.**
>
> **A4**: We have compared the two popular FL methods, Moon[1] and Fedopt[2]. We conducted experiments on the ImageNette and ImageNet100 datasets, considering a scenario with 50 clients under non-IID settings (beta=0.5). To the best of our knowledge, there is currently no federated learning method that surpasses centralized training. However, our proposed method even outperforms centralized trained models in many scenarios (see Table 1 in main text). Therefore, as shown in this table, our method still outperforms other federated learning approaches.
>
> | Method | FedAvg | FedOpt | Moon | Ours (one-shot) | Ours (5-round) |
> |:----------------------:|:------:|:------:|:-----:|:---------------:|:--------------:|
>  | ImageNette (beta=0.5) | 72.01 | 73.21 | 74.27 | 85.21 | 93.80 |
> | ImageNet100 (beta=0.5) | 40.13 | 41.25 | 41.43 | 48.31 | 72.67 |
>
> [1]Li, Qinbin, Bingsheng He, and Dawn Song. "Model-contrastive federated learning." Proceedings of the IEEE/CVF conference on computer vision and pattern recognition. 2021.
>
> [2]Reddi, Sashank J., et al. "Adaptive Federated Optimization." International Conference on Learning Representations. 2020.
>
> **Q5: Table 2 shows the proposed method outperforming centralized learning significantly; a thorough explanation of this phenomenon is warranted.**
>
> **A5**: This is because our method synthesizes a balanced dataset using all collected prompts during the first round of communication. We then pretrain a "well-initialized" model on this dataset. Once we have this well-initialized model, several rounds of communication can quickly bring the model to a good performance. In the first table, we present the results of directly loading a pretrained model on ImageNet. It can be observed that directly loading a pretrained model on ImageNet reduces the gap between our method and FedAvg. This is because the pretrained model provides a good starting point. However, training a pretrained model on ImageNet requires a significant computational cost on a dataset of 1.3M samples. In contrast, our method only requires training on a small amount of synthesized data to provide a well-initialized model, hence achieving better performance than models trained in a centralized manner.

---

> ### Author Response · Authors · 2023-11-20
> **A friendly reminder that the discussion stage will be closed in 2 days**
>
> Dear Reviewer,
>
> Thank you once again for your valuable comments. As the discussion stage is coming to a close in 2 days, we kindly request your feedback on whether our response adequately addresses your concerns. We would greatly appreciate any additional feedback you may have.
>
> Thank you in advance!
>
> Kind regards,
>
> Authors

---

> > ### Comment · Reviewer_ZNb2 · 2023-11-22
> > **Experimentation with other datasets and the experimental results**
> >
> > I would appreciate the rebuttal of the authors. However, I have two major concerns.
> > 1. The authors mentioned that they have limited time and cannot conduct NLP tasks. However, the authors argue that it is easy to apply the approach to NLP tasks.
> > 2. It is hard to understand how the FL results could be better than centralized approches. I wonder if the authors could explain the indepth reason.

---

> > > ### Author Response · Authors · 2023-11-22
> > > **Thanks for Response**
> > >
> > > Thank you for your response!
> > >
> > > 1. Since our method focuses on using foundation models in FL, it should be relatively easy to adapt to NLP tasks as well, where foundation models are also used for data synthesis [1,2].
> > >
> > > [1] Yue, Xiang, et al. "Synthetic text generation with differential privacy: A simple and practical recipe." arXiv preprint arXiv:2210.14348 (2022).
> > >
> > > [2] Veselovsky, Veniamin, et al. "Generating Faithful Synthetic Data with Large Language Models: A Case Study in Computational Social Science." arXiv preprint arXiv:2305.15041 (2023).
> > >
> > > 2. Why is it not possible for our method to perform better than centralized training on certain datasets?  Consider this: when you pretrain a model on ImageNet and then use it for certain downstream tasks, it often performs better than training from scratch, such as on the Cars and CUB-200 datasets. Our method provides a well-initialized model in the first round, hence achieving better performance than models trained in a centralized manner.
> > >
> > > Please feel free to let us know if there are any further questions.

---

### Official Review · Reviewer_wnsW · 2023-10-30

**Soundness:** 2 fair
**Presentation:** 4 excellent
**Contribution:** 2 fair
**Rating:** 5
**Confidence:** 4

**Summary:**

- The main idea of the paper is to use prompts to “summarize” the client-side data in federated learning. These prompts are then sent to the central server and fed to a foundation generative model, with the hope that the generated data distribution is close to the client data distribution.
- With this idea, federated learning can be made one-round or few-round to drastically reduce communication costs, where clients can just send over the prompts one-shot to the server as the prompts and labels require very little communication.
- The paper then evaluates on several natural image datasets (subsets from ImageNet) and show that the proposed technique can match FedAvg in performance.
- The paper also performs some privacy analysis and shows that by transmitting prompts instead gradients/model updates/data, the membership inference attack success drops significantly.

**Strengths:**

- The proposed approach is interesting and novel to my understanding. Assuming the client data distributions can be well captured by the foundation generative model, the proposed technique can clear benefits in simplicity and reducing communication costs.
- Putting aside the underlying assumptions of the proposed techniques (see weaknesses), the paper is overall well-executed in terms of the diversity of the experiments and visualizations.
- The paper is generally well-written and easy-to-follow.

**Weaknesses:**

[W1] The main weakness of the proposed method is the underlying assumption that client data can, in fact, be generated by foundational models. This sound obvious but is key to the applicability of the proposed approach in practice. To put it bluntly, is the proposed solution searching for a problem?

1. Settings where FL is helpful—such as medical images across hospitals [1], user-generated text across mobile phones [2]—are often where the data distributions aren’t covered by the pre-training data of foundational models. The datasets used by the experiments are all natural image datasets (ImageNette, ImageFruit, etc.), which can be well-represented in the pre-training dataset of foundation generative models. I would appreciate results on non-natural image datasets.
2. In particular, if we consider horizontal FL settings (as with the paper), the server may even know about the possible classes / labels (e.g. federating binary classifiers) without communicating to the clients, in which case the “class-level prompts” may not be needed at all since the server can just generate images by itself.

[W2]  More broadly, the threat model of the paper may need to be defined more clearly.

- What exactly is client privacy in this case? Can the client data be still considered “private” if you could already generate them with public foundation models (see also [3])? Does the privacy of the data lie in the pixels, or simply the description of the pixels?
- In many cases, the descriptions of the images can already be leaking privacy. If we apply the proposed method to cross-device federated learning on user’s photo data, the server could already learn a lot about the user data distribution and preferences. For example, following Sec 5.4 and Figure 6, knowing that a user have lots of golf photos (without knowing the pixels of the photos) already allows the FL service provider (e.g. Google) to sell targeted ads.

[1] FLamby: Datasets and Benchmarks for Cross-Silo Federated Learning in Realistic Healthcare Settings. NeurIPS 2022 Datasets and Benchmark.  https://arxiv.org/abs/2210.04620
[2] https://research.google/pubs/pub47586/
[3] Considerations for Differentially Private Learning with Large-Scale Public Pretraining. https://arxiv.org/pdf/2212.06470.pdf

**Questions:**

- [Intro section] Why exactly does the proposed method provide robustness to data heterogeneity? Heterogeneity can still surface in the (instance-level) client prompts and subsequently the generated images.
- Minor comment: consider using different citation commands `\citet` , `\cite`, etc. in LaTeX to make the formatting of the in-text references consistent.

---

> ### Author Response · Authors · 2023-11-17
> **Response to wnsW [1/2]**
>
> We thank Reviewer wnsW for the valuable feedback and insightful comments. Here, we answer your questions and provide more experimental evidence.
>
> **Q1: The datasets used by the experiments are all natural image datasets (ImageNette, ImageFruit, etc.), which can be well-represented in the pre-training dataset of foundation generative models. I would appreciate results on non-natural image datasets.**
>
> **A1:** Although the pretraining dataset and the private dataset may exhibit some domain similarities (e.g., both may contain common real-world scenes), the tasks of lmageSquawk (fine-grained bird classification) and QuickDraw (non-realistic domain) in DomainNet that we demonstrate in our experiments are inherently challenging. Training a model solely on synthetic data generated by foundation models to achieve high accuracy on the ImageNet or DomainNet test sets is a non-trivial task.
> To further validate the effectiveness of our method, we conducted experiments on several fine-grained image classification datasets, including CUB, Cars, and the satellite image dataset EuroSAT. As the official EuroSAT dataset did not provide predefined training and testing splits, we performed a split in an 8:2 ratio. The size of fine-grained recognition datasets is typically smaller compared to general image classification datasets. In previous work, a common practice is to utilize a pretrained model that has been trained on the ImageNet dataset. In this study, we present two approaches: training the model from scratch and loading a pretrained ResNet34 model. As shown in the table, our method achieves excellent performance even in these challenging domains. Additionally, in the cross-silo federated learning scenario, when clients have strong computational capabilities, one can simply finetune the foundation models on these domains, achieving better performance than normal federated learning methods. We have added these results in the appendix.
>
>
> | |               |         |       |             |              ||  |             |       |  |
> |:---------------:|:-------------:|:-------:|:------------:|:-----------:|:------------:|:---------------:|:--------------:|:-----------:|:-----:|:-----------:|
> |   Prompt type   | Training type | Dataset | FedAvg,$\beta=0.01$ |FedAvg, $\beta=0.5$ | FedAvg (IID) |    Ours (one-shot)              |  Ours (5-round) ,$\beta=0.01$  |Ours (5-round),$\beta=0.5$ |  Ours (5-round),IID  |        Centralized     |
> |     instance    |    scratch    | CUB-200 |     35.04    |    36.61    |     36.62    |      44.17      |      64.53     |    69.19    | 71.01 |    48.31    |
> |     instance    |   pretrain    | CUB-200 |     78.98    |    79.08    |     78.48    |      54.02      |      75.13     |    78.96    | 80.72 |    81.77    |
> |      class      |    scratch    | CUB-200 |     35.04    |    36.61    |     36.62    |      45.34      |      67.66     |     71.9    | 73.33 |    48.32    |
> |      class      |   pretrain    | CUB-200 |     78.98    |    79.08    |     78.48    |      52.73      |      74.68     |     78.7    | 80.32 |    81.77    |
> |      class      |    scratch    |   Cars  |     55.18    |    42.43    |     44.48    |      54.48      |      83.31     |    87.22    | 88.07 |    64.72    |
> |      class      |   pretrain    |   Cars  |     87.71    |    88.91    |     88.96    |      60.55      |      87.31     |    90.05    | 90.73 |    91.21    |
> |      class      |    scratch    | EuroSAT |     43.94    |    74.48    |     84.87    |      38.37      |      37.59     |    82.94    | 91.01 |    94.31    |
>
> **Q2: the “class-level prompts” may not be needed at all since the server can just generate images by itself.**
>
> **A2**: Yes, if the server-side has knowledge of the specific labels for the classification task, it can generate them directly. However, class-level is just a simple case. We propose the instance-level approach to address more complex domains, where client-side customized prompt generation is more advantageous in improving the performance of the overall model.
>
> **Q3: More broadly, the threat model of the paper may need to be defined more clearly.**
>
> **A3**: threat model: In traditional federated learning schemes that transmit model parameters/gradients, attackers can launch various attacks once they obtain the parameters/gradients, such as membership inference attacks, adversarial example attacks, and model inversion. In contrast, our approach significantly reduces potential security and privacy risks because users only transmit prompts in the first round of communication. To the best of our knowledge, there is no research indicating that using prompts alone can perfectly reconstruct private data. Therefore, our approach is more secure and privacy-preserving compared to FedAvg.

---

> ### Author Response · Authors · 2023-11-17
> **Response to wnsW [2/2]**
>
> **Q4: What exactly is client privacy in this case? Can the client data be still considered “private” if you could already generate them with public foundation models.**
>
> **A4**: client privacy: In this paper, similar to differential privacy, we primarily focus on individual privacy, as it is more challenging for attackers. For instance, Attack A perfectly targets a known subset of 0.1% of users in a client, but succeeds with a random 50% chance on the rest. Attack B succeeds with a 50.05% probability on any given user in a client. On average, these two attacks have the same attack success rate. However, the second attack is practically useless, while the first attack is much more powerful in the real-world. This is precisely what LiRA[1] emphasizes, as it evaluates the privacy attack by computing their true-positive rate at very low (e.g., ≤ 0.1%) false-positive rates (as illustrated in our experimental results in Figure 8), demonstrating that our method can better defend against privacy attacks.
>
> [1] Carlini, Nicholas, et al. "Membership inference attacks from first principles." 2022 IEEE Symposium on Security and Privacy (SP). IEEE, 2022.
>
> **Q5: In many cases, the descriptions of the images can already be leaking privacy.**
>
> **A5**:This could be the difference between individual privacy and group privacy. The majority of the current papers on data protection focuses on the individual ‘user,’ or ‘data subject,’ who’s right to privacy will grow exponentially with the enforcement of the GDPR (General Data Protection Regulation). However, group privacy is not mentioned in the GDPR , which is not well-defined. Also, if the server knows some of the user data distribution means potential privacy risks, our proposed method does not introduce additional risk in this regard. This is because in traditional gradient/parameter-based methods, using model inversion, the server can still infer this information[1,2]. But for individual privacy, this information won't increase the leakage of membership in private data.
>
> [1] Geiping, Jonas, et al. "Inverting gradients-how easy is it to break privacy in federated learning?." Advances in Neural Information Processing Systems 33 (2020): 16937-16947.
>
> [2] Hatamizadeh, Ali, et al. "Do gradient inversion attacks make federated learning unsafe?." IEEE Transactions on Medical Imaging (2023).
>
> **Q6:Why exactly does the proposed method provide robustness to data heterogeneity? Heterogeneity can still surface in the (instance-level) client prompts and subsequently the generated images.**
>
> **A6**: For one-shot Federated Learning (FL), regardless of the extreme data distributions among different clients, the server can always collect prompts corresponding to all the data, thus obtaining a balanced synthetic dataset on the server. Therefore, compared to FedAvg, our method is not sensitive to data heterogeneity in the first round of communication. In the subsequent model updates, the clients are still affected by non-IID data. However, due to the well-trained initial model obtained in the first round of communication, only a few rounds of communication are needed for local updates, making it more robust to data heterogeneity. As shown in Table 1 in the main text, our method exhibits significantly smaller gaps compared to FedAvg under different non-IID scenarios.
>
> **Q7: Minor comment: consider using different citation commands \citet , \cite**
>
> **A7:** Thanks for pointing this out. we will check it in the updated version.

---

> ### Author Response · Authors · 2023-11-20
> **A friendly reminder that the discussion stage will be closed in 2 days**
>
> Dear Reviewer,
>
> Thank you once again for your valuable comments. As the discussion stage is coming to a close in 2 days, we kindly request your feedback on whether our response adequately addresses your concerns. We would greatly appreciate any additional feedback you may have.
>
> Thank you in advance!
>
> Kind regards,
>
> Authors

---

> ### Comment · Reviewer_wnsW · 2023-11-21
> **Response to author rebuttal**
>
> ###
>
> I appreciate the authors for providing a rebuttal.
>
> - A1: I appreciate the authors for putting efforts into the new results. I also appreciate pointing to the results on QuickDraw. However, my concern is not fully addressed since the datasets “CUB-200” (natural images of birds) and “Cars” (natural images of cars) are very much still in-distribution for the pre-trained generative vision models.
> - A2: The authors responded to my question by pointing to the use of instance-level prompts, but this didn’t quite address my concern that the significance of the class-level prompts is a bit overclaimed. Considering the default implementation of your experiments uses class-level prompts (page 6), I would suggest clearly spelling out the assumptions and weaknesses of class-level prompts in the updated paper.
> - A4/A5:
>     - (For clarity, the following discussions apply to “instance-level” prompts)
>     - By explaining the LiRA paper on membership inference in A4, the authors imply that the paper cares about instance-level privacy — i.e. image-level privacy, where an attacker cannot confidently tell whether one image is or isn’t used for training.
>     - I’m definitely okay with the **privacy granularity** in this case; what I’m uncertain about (with Q5) is whether **all the information contain within a single example (i.e. image-label pair)** is protected.
>     - A5 does not quite address my question. I do not agree that this is the difference between “group privacy” vs “individual privacy”; rather it is that the instance-level prompts have provided **side channels into learning about the information of a single image.**
>     - Consider running local, image-level DP-SGD on a client when participating in a vanilla FedAvg task. All the information corresponding to a single example (pixel values and labels) are protected behind the “privacy barrier” since privatized gradients are applied to the model. In contrast, instance-level prompts would leak information about the pixel values, and thus do not really satisfy instance-level privacy in the sense of “attacker not being able to tell whether an image is used for training”. I do acknowledge however that there is value in providing empirical privacy of the pixel values.
> - A6: Thanks for the clarification that the server can select/curate prompts to essentially manually mitigate the data heterogeneity. I would suggest highlighting this in the updated version.
>
> Overall, the technique proposed in the paper is interesting, though I feel the assumptions on client data distributions and privacy claims are too strong. Having read through other reviewers’ comments, I’m keeping my rating at 5.

---

> > ### Author Response · Authors · 2023-11-21
> > **Response**
> >
> > Thank you for your valuable time.
> >
> > A1: So `you just ignore our results on QuickDraw and EuroSAT`, which also performs much better compared with traditional FL methods and refute your claim.
> >
> > A2: Thanks for your suggestion. The default implementation of our experiments utilizes class-level prompts, as they have shown to provide sufficiently good performance and effectively protect privacy. The choice of training method depends on the specific use case. It is important to note that there is no perfect approach in data privacy, as no method can guarantee zero information leakage while achieving a perfect model (`no free lunch in privacy`).
> >
> > A4/A5:
> > - `Can you find any method in FL that outperforms our method and efficiently defends against LiRA attack` (the most powerful membership inference attack)? To the best of our knowledge, no other method has been shown to achieve such performance.
> >
> > - I understand your concern about potential risks associated with prompts. Consider the following: `Can you reconstruct any private data using only these prompts`?  This task is extraordinarily challenging, even in the complete white-box setting, where the prompt cannot perfectly reconstruct the private data. Not to mention, our scenario generation model has never been trained on private data.
> >
> > A6: Thank you for acknowledging the robustness of our method in handling non-IID data.
> >
> > We aim to motivate researchers to consider the effective integration of foundation models for downstream tasks in federated learning through our approach. Additionally, we encourage researchers to explore and identify viable attack strategies to demonstrate the method's potential lack of privacy preservation, either **theoretically or experimentally**.

---

### Official Review · Reviewer_HdAm · 2023-11-01

**Soundness:** 3 good
**Presentation:** 3 good
**Contribution:** 3 good
**Rating:** 6
**Confidence:** 3

**Summary:**

This work introduces a novel federated learning framework called Federated Generative Learning, which addresses the inefficiency and privacy issues of existing solutions that transmit features, parameters, or gradients between clients and servers. In this framework, clients generate text prompts tailored to their local data and send them to the server, where informative training data is synthesized using stable diffusion. This approach offers enhanced communication efficiency, significant performance gains, and improved privacy protection, as demonstrated through extensive experiments on ImageNet and DomainNet datasets.

**Strengths:**

- This work proposes a novel learning framework to train local data without accessing the raw data directly.

- communication of prompts instead of model parameters addresses several issues of existing federated learning frameworks; high communication cost and potential privacy threats by attackers.

**Weaknesses:**

- The proposed method may be highly dependent on the performance of both diffusion models and visual-captioning models.
  - An ablation study of varying the foundation models is needed.

- In a similar vein, the local training dataset should be unseen for pertaining foundation models and should be more difficult than ImageNet which is a standard image classification dataset. As mentioned in the Introduction section, the local training data are more likely to be privacy sensitive, so they are more likely to be unseen or not contained for pre-training foundation models such as BLIPv2 and Stable Diffusion. Evaluation on ImageNet or DomainNet implicitly uses the assumption that local data have a similar or subset domain to the pretraining dataset of foundation models, which are publically accessible or have no privacy issue.

- Clients in federated learning are often assumed to have limited capacity in memory or computation. Generating prompts using a large visual captioning model in each client is impractical.

**Questions:**

- The quality of synthetic data could be highly different according to domain discrepancy between the local training data and the pretraining data for the foundation model. Instead of using standard image classification datasets, does the proposed method work for federated learning on fine-grained classification such as CUB-200, Cars, and medical image datasets?

---

> ### Author Response · Authors · 2023-11-17
> **Response [1 / 2]**
>
> We thank Reviewer HdAm for your valuable feedback and constructive comments. We have carefully answered all your questions and added extra experiment results in the following.
>
> **Q1: The proposed method may be highly dependent on the performance of both diffusion models and visual-captioning models. An ablation study of varying the foundation models is needed.**
>
> **A1**: Thanks for this insightful point. To investigate the impact of various generative models on the results, we followed the setting in [1]. Our experiments primarily focus on three prevalent conditional diffusion models: DiT[2], GLIDE[3], and Stable Diffusion. We use these off-the-shelf models to generate synthetic images. Specifically, for GLIDE and Stable Diffusion, the prompt was configured as "a photo of {label name}, real-world images, high resolution." For DiT, the input comprised the label ID corresponding to the ImageNet1k dataset. The images synthesized by DiT and GLIDE are of dimensions 256x256, whereas those produced by Stable Diffusion are of dimensions 512x512. As shown in the following table, even when we vary the foundation models used in our method, FGL consistently outperforms FedAvg by a significant margin. This observation serves as evidence for the generality of our approach. We have added these results in the Appendix A.4.3.
>
> |           Method          | one-shot | 5-round, $\beta$=0.01 | 5-round, $\beta$=0.5 |    IID   |
> |:-------------------------:|:--------:|:------------------:|:-----------------:|:--------:|
> | Ours w/  Stable Diffusion | **85.2** |      **82.8**      |       **94.1**      | **95.6** |
> |       Ours w/ Glide       |    79.0    |        76.2        |        89.4       |   89.4   |
> |        Ours w/ Dit        |   76.2   |        74.6        |        90.2       |   92.8   |
> |     FedAvg (120-round)    |     -    |        51.6        |        75.1       |   79.2   |
>
>
> [1] Li, Zheng, et al. "Is Synthetic Data From Diffusion Models Ready for Knowledge Distillation?." arXiv preprint arXiv:2305.12954 (2023).
>
> [2] Nichol, Alex, et al. "Glide: Towards photorealistic image generation and editing with text-guided diffusion models." arXiv preprint arXiv:2112.10741 (2021).
>
> [3]Peebles, William, and Saining Xie. "Scalable diffusion models with transformers." Proceedings of the IEEE/CVF International Conference on Computer Vision. 2023.
>
>
> **Q2: Clients in federated learning are often assumed to have limited capacity in memory or computation.**
>
>
> **A2**:
> - First, compared to FedAvg, our method introduces only one additional operation on the client side, i.e., prompt generation, which involves only forward propagation and does not impose significant computational costs. All computational operations are executed on the server side during the initial communication. The server trains a model with an excellent initial state, and subsequently, the client performs regular model updates, which means no additional cost compared with FedAvg.
> - Secondly, our method is particularly well-suited for cross-silo FL, where the clients represent organizations or companies. In this context, the number of clients is typically small, but they possess substantial computational resources. Furthermore, this scenario emphasizes the importance of protecting clients' local data from potential leaks, which constitutes a significant contribution of our approach towards preserving privacy.

---

> ### Author Response · Authors · 2023-11-17
> **Response [2 / 2]**
>
> **Q3.Evaluation on ImageNet or DomainNet implicitly uses the assumption that local data have a similar or subset domain to the pretraining dataset of foundation models, which are publically accessible or have no privacy issue.**
>
> **A3**: Thanks for pointing this out. Here, we would like to address this from three aspects:
> - Although the pretraining dataset and private dataset may have some domain similarities (e.g., both may contain common real-world scenes), the tasks of lmageSquawk (fine-grained bird classification) and QuickDraw (non-realistic domain) in DomainNet that we show in our experiments are challenging. It is a non-trivial task to train a model only using synthetic data generated by foundation models to achieve good accuracy on the ImageNet or DomainNet test sets.
>
> - Furthermore, even when there are some domain similarities between the pretraining dataset and private dataset, does it mean that there is no need to discuss the privacy risk? Definitely not! Let's consider a scenario where a public dataset contains various images of cats, while a private dataset contains personal images of cats belonging to individual users. Although both datasets involve images of cats, the private dataset may contain users' personal information, such as their family photos or addresses. Therefore, even if the two datasets are similar in some aspects, the private data still carries privacy risks and needs to be properly protected. Taking Membership Inference Attack (MIA) as an example, consider an adversary that wants to probe a ML model to test membership of an individual's data in the model's training data. In this scenario, an adversary is more likely to have access to some representative images of the target individual, but not necessarily the ones used for training the model. As shown in Figure 8, we implemented the state-of-the-art LiRA algorithm in MIA. The experimental results demonstrate that our approach ensures the protection of sensitive information of the members in the clients' data (since the model training process has never been exposed to any private data). In contrast, traditional federated learning methods directly train on private data, posing a high risk of exposing the sensitive information of the members in the clients' data (i.e., for certain private data samples, attackers have a high confidence in identifying the client from which the sample originates). To the best of our knowledge, prior to our proposed approach, no one has put forth a training paradigm that effectively defends against LiRA while concurrently modeling utility (i.e., achieving high test accuracy).
>
> - Finally, even for particularly challenging domains such as remote sensing images or fine-grained classification datasets, our method can easily adapt to these scenarios. We conducted experiments on several fine-grained image classification datasets, namely CUB-200, Stanford Cars, and also the satellite image dataset EuroSAT. CUB-200 is a challenging dataset consisting of 200 bird species, while Stanford Cars contains 16,185 images belonging to 196 classes of cars. See more details in the `Appendix A.4.2`.
>
> | |               |         |       |             |              ||  |             |       |  |
> |:---------------:|:-------------:|:-------:|:------------:|:-----------:|:------------:|:---------------:|:--------------:|:-----------:|:-----:|:-----------:|
> |   Prompt type   | Training type | Dataset | FedAvg,$\beta=0.01$ |FedAvg, $\beta=0.5$ | FedAvg (IID) |    Ours (one-shot)              |  Ours (5-round) ,$\beta=0.01$  |Ours (5-round),$\beta=0.5$ |  Ours (5-round),IID  |        Centralized     |
> |     instance    |    scratch    | CUB-200 |     35.04    |    36.61    |     36.62    |      44.17      |      64.53     |    69.19    | 71.01 |    48.31    |
> |     instance    |   pretrain    | CUB-200 |     78.98    |    79.08    |     78.48    |      54.02      |      75.13     |    78.96    | 80.72 |    81.77    |
> |      class      |    scratch    | CUB-200 |     35.04    |    36.61    |     36.62    |      45.34      |      67.66     |     71.9    | 73.33 |    48.32    |
> |      class      |   pretrain    | CUB-200 |     78.98    |    79.08    |     78.48    |      52.73      |      74.68     |     78.7    | 80.32 |    81.77    |
> |      class      |    scratch    |   Cars  |     55.18    |    42.43    |     44.48    |      54.48      |      83.31     |    87.22    | 88.07 |    64.72    |
> |      class      |   pretrain    |   Cars  |     87.71    |    88.91    |     88.96    |      60.55      |      87.31     |    90.05    | 90.73 |    91.21    |
> |      class      |    scratch    | EuroSAT |     43.94    |    74.48    |     84.87    |      38.37      |      37.59     |    82.94    | 91.01 |    94.31    |
>
> **Q4: does the proposed method work for federated learning on fine-grained classification such as CUB-200, Cars, and medical image datasets?**
>
> **A4**: Please refer to the table in A3.

---

> ### Author Response · Authors · 2023-11-20
> **A friendly reminder that the discussion stage will be closed in 2 days**
>
> Dear Reviewer,
>
> Thank you once again for your valuable comments. As the discussion stage is coming to a close in 2 days, we kindly request your feedback on whether our response adequately addresses your concerns. We would greatly appreciate any additional feedback you may have.
>
> Thank you in advance!
>
> Kind regards,
>
> Authors

---

### Author Response · Authors · 2023-11-17
**Summary**

Dear Reviewers and ACs,

Thank you all for your insightful reviews and constructive comments on our manuscript. We greatly appreciate your feedback, which has helped us improve our work. We have carefully considered all the suggestions and made the following changes:

1. We have included three additional challenging datasets from different domains: CUB-200 and Stanford Cars, which are fine-grained image classification datasets, and the EuroSAT satellite image dataset, which is known to be more difficult to generate. We have conducted experiments on these datasets to demonstrate the effectiveness of our method in various scenarios.

2. To showcase the versatility of our proposed approach, we have employed diverse Generative models. By doing so, we aim to demonstrate that our method is not limited to a specific model but can be applied to different models with similar success.

3. In order to provide more robust evidence of the effectiveness and scalability of our proposed method, we have conducted experiments with an increased number of clients. Specifically, we have included experiments with 50 and 100 clients, which further support our findings and demonstrate the scalability of our approach.

4. We have included extensive discussions on the security and privacy aspects of our proposed method. We believe that addressing these concerns is crucial, and we have provided thorough explanations and considerations to ensure the privacy and security of the data used in our experiments.

Thank you once again for your valuable feedback.

Best Regards,
All authors.

---

### Meta-Review · Area_Chair_8sJH · 2023-12-05

**Metareview:**

This paper presented an interesting approach to federated learning that doesn't require each client to send the parameters or the gradients to the server but only sends a text prompt describing the client data. The server can then synthesize training data using the received prompts and train.

While the reviewers appreciated the idea, a number of concerns were raised. Some of these were related to the fact the datasets used in the experiments could have been present in the pre-training dataset of the foundation model, and applicability of the method to datasets more sophisticated than ImageNet.

The authors provided a detailed rebuttal and the paper was discussed. However, apart from one reviewer who marginally leaned towards acceptance (though still had some concerns), the other three reviewers maintained their original assessment and their concerns remained.

In the end, after taking into account the reviews, the discussion, and my own reading of the paper, the paper falls short of the acceptance threshold. Although the authors did respond to the reviewers' concerns with additional experimental results, the paper in its current form still does not seem ready for publication. It is advised that the authors properly incorporate the concerns raised in the reviews and submit the work at another venue.

**Justification For Why Not Higher Score:**

The concerns of the reviewers still persisted after the author response and discussion

**Justification For Why Not Lower Score:**

N/A

---

### Decision · Program_Chairs · 2024-01-16

Reject